# On current-squared flows and ModMax theories

Christian Ferko[1*], Liam Smith[2†] and Gabriele Tartaglino-Mazzucchelli[2‡]

**1** Center for Quantum Mathematics and Physics (QMAP), Department of Physics & Astronomy, University of California, Davis, CA 95616, USA
**2** School of Mathematics and Physics, University of Queensland, St Lucia, Brisbane, Queensland 4072, Australia

★ caferko@ucdavis.edu, † liam.smith1@uq.net.au, ‡ g.tartaglino-mazzucchelli@uq.edu.au

## Abstract

We show that the recently introduced ModMax theory of electrodynamics and its Born-Infeld-like generalization are related by a flow equation driven by a quadratic combination of stress-energy tensors. The operator associated to this flow is a $4d$ analogue of the $T\overline{T}$ deformation in two dimensions. This result generalizes the observation that the ordinary Born-Infeld Lagrangian is related to the free Maxwell theory by a current-squared flow. As in that case, we show that no analogous relationship holds in any other dimension besides $d = 4$. We also demonstrate that the $\mathcal{N} = 1$ supersymmetric version of the ModMax-Born-Infeld theory obeys a related supercurrent-squared flow which is formulated directly in $\mathcal{N} = 1$ superspace.


# 1   Introduction

Since 2016 there has been a wide range of activities surrounding the study of $d = 2$ quantum field theories deformed by the irrelevant $T\overline{T}$ operator [1–3]. Such operator is defined as the determinant of the stress-energy tensor, $O_{T\overline{T}} \propto \det T_{\mu\nu}$, which in $d = 2$ is equivalent to the following quadratic combination

$$O_{T\overline{T}} \propto \left( T^{\mu\nu} T_{\mu\nu} - \Theta^2 \right), \qquad \Theta := T^{\mu}_{\mu}. \tag{1.1}$$

Despite being irrelevant, the local operator $O_{T\overline{T}}$ proves to be quantum mechanically well defined [1, 2] and to preserve many of the symmetries of the seed theory, including integrability [2, 4, 5], and supersymmetry [6–9]. By now the field of research surrounding $T\overline{T}$-like deformations contains a large body of literature which we will not attempt to review in detail here. Instead we refer the reader to [10] for a pedagogical introduction to the subject.

Within this context, the main focus of this paper concerns classical Lagrangian flows triggered by current-squared $T\overline{T}$-like operators. The $T\overline{T}$ deformation of a two-dimensional theory leads to a classical flow equation for the deformed Lagrangian $\mathcal{L}_\lambda$ of the form

$$\frac{\partial}{\partial \lambda} \mathcal{L}_\lambda = -\frac{1}{8} O_{T\overline{T}} \propto \det \left( T_{\mu\nu}[\mathcal{L}_\lambda] \right), \tag{1.2}$$

where $T_{\mu\nu}[\mathcal{L}_\lambda]$ is the stress-energy tensor for the deformed theory at value $\lambda$ of the flow parameter. Solving this type of flow equation proves, on the one hand, to be a fairly involved task even for classical systems, and in the last few years various direct, geometric, and string theory inspired techniques have been developed to tackle this problem [3, 6, 7, 11–19]. On the other hand, the solutions to such flows lead to surprising and remarkable results. The simplest example, that was considered for the first time in [3], is the deformation of the Lagrangian of a free real scalar field in $d = 2$ dimensions. The undeformed Lagrangian is

$$\mathcal{L}_0 = \frac{1}{2} \partial^\mu \phi \partial_\mu \phi. \tag{1.3}$$

The deformed Lagrangian $\mathcal{L}_\Lambda$ satisfying (1.2) was shown to be [3] (see also [11])

$$\mathcal{L}_\lambda = -\frac{1}{2\lambda} + \frac{1}{2\lambda} \sqrt{1 + 2\lambda \partial^\mu \phi \partial_\mu \phi}, \tag{1.4}$$

which is the gauge-fixed Nambu-Goto Lagrangian for a string with tension determined by $\lambda$. This simple result is one of the many links that $T\overline{T}$ deformations have found with string theory, see e.g. [6, 14, 18, 20–24], and shows how these deformations can be used to shed new light on the realm of non-local quantum field theories. The result (1.4) was also extended to the supersymmetric case, where $\mathcal{N} = (0, 1), (1, 1), (0, 2)$ and $(2, 2)$ supersymmetric extensions of the Nambu-Goto string were proven to be $T\overline{T}$-flows [6–9, 15, 25]. Such proofs made use of

manifestly supersymmetric forms of $T\overline{T}$ formulated in superspace in terms of supercurrent-squared operators [6–9].

Extensions of the Lagrangian flow (1.2) in terms of operators defined by squared combinations of the stress-energy tensor have been considered also in $d > 2$.[1] One notable example is the proposal of [27, 28] that arises from an holographic interpretation of $T\overline{T}$-like deformations in $d \geq 2$. Another very surprising and inspiring example is the flow equation that has been discovered in [13] for the Maxwell-Born-Infeld theory. Its Lagrangian

$$\mathcal{L}_{\mathrm{BI}} = \frac{1}{\alpha^2}\left\{ 1 - \sqrt{1 + \frac{\alpha^2}{2}F^2 - \frac{\alpha^4}{16}(F\tilde{F})^2} \right\},$$ (1.5a)

was in fact shown to be a deformation of the free Maxwell Lagrangian as follows:

$$\frac{\partial \mathcal{L}_{\mathrm{BI}}}{\partial \alpha^2} = \frac{1}{8}\left(T^{\mu\nu}T_{\mu\nu} - \frac{1}{2}\Theta^2\right), \qquad \mathcal{L}_{\mathrm{BI}}\big|_{\alpha^2=0} = -\frac{1}{4}F^2 = \mathcal{L}_{\mathrm{Maxwell}}.$$ (1.5b)

Extensions of this result were considered in [25, 29, 30]. For instance, a manifestly supersymmetric extension of the flow equation (1.5b) was proven in [25] to hold for the 4$d$, $\mathcal{N} = 1$ supersymmetric Maxwell-Born-Infeld theory proposed by Bagger and Galperin in [31]. The (supersymmetric) Born-Infeld theory is of great importance due to its role in the low-energy, effective description of brane systems in string theory. From this point of view, the flow (1.5b) reads as a 4$d$ extension of the 2$d$ Nambu-Goto case and raises questions about whether current-squared flows might be a universal feature of string theory yet to be uncovered.

One of the well-known features that characterises Maxwell theory, together with its Born-Infeld extension, is invariance under electro-magnetic duality, a property which is also shared by its Bagger-Galperin supersymmetric extension. This U(1) duality symmetry can be thought of as a phase rotation of a complex combination of the field strength $F_{\mu\nu}$ and its dual $\widetilde{F}^{\mu\nu} = \frac{1}{2}\epsilon^{\mu\nu\rho\sigma}F_{\rho\sigma}$:

$$F_{\mu\nu} + \mathrm{i}\widetilde{F}_{\mu\nu} \longrightarrow e^{\mathrm{i}\theta}\left(F_{\mu\nu} + \mathrm{i}\widetilde{F}_{\mu\nu}\right).$$ (1.6)

It is natural to ask whether there are other theories of electromagnetism which also exhibit such eletro-magnetic symmetry (1.6). In this context, recently it was discovered in [32] (see also [33]) that there is a unique one-parameter family of Lorentz invariant modifications of the Maxwell Lagrangian in $d = 4$ which preserve both duality invariance and conformal symmetry. This unique deformation is called the Modified Maxwell (or ModMax) theory and is described by the Lagrangian

$$\mathcal{L}_{\mathrm{ModMax}} = -\frac{1}{4}\cosh(\gamma)F^2 + \frac{1}{4}\sinh(\gamma)\sqrt{(F^2)^2 + (F\widetilde{F})^2}.$$ (1.7)

Here $\gamma$ is a dimensionless real parameter that controls the deformation; when $\gamma = 0$, the Lagrangian (1.7) reduces to the usual Maxwell theory. Since the equations of motion for Maxwell theory are duality invariant, and because the combination under the square root in (1.7) is proportional to $z_{\mu\nu}z^{\mu\nu}\overline{z}^{\rho\sigma}\overline{z}_{\rho\sigma}$ where $z_{\mu\nu} = F_{\mu\nu} + \mathrm{i}\widetilde{F}_{\mu\nu}$, the ModMax theory is also invariant under U(1) duality rotations (1.6). Note that study of duality invariant models, with and without supersymmetry, has a very long history. We refer the reader to the following (incomplete) list of papers and references therein [32–49]. For a pedagogical introduction to theories of non-linear electrodynamics such as ModMax, see [50].

---

[1]Unlike the $d = 2$ case, it is not known whether such classical flows correspond to well-defined operators at the quantum mechanical level. Understanding the quantum properties of $T^2$ flows in $d > 2$, perhaps with additional assumptions such as maximal supersymmetry, remains an important open question. Related interesting developments in this direction were obtained for $d = 4$, $\mathcal{N} = 4$ SYM in [26].

Due to the presence of the square root in (1.7), one is tempted to compare ModMax with the Maxwell-Born-Infeld theory (1.5a). However, it is clear that the ModMax theory of electrodynamics is qualitatively quite different from the Born-Infeld theory and not only because one is conformal and the other is not. Although both Lagrangians involve square roots, the Born-Infeld Lagrangian (1.5a) can be Taylor expanded around small field strength to yield the Maxwell Lagrangian plus an infinite series of higher-derivative corrections. But since the square root appearing in the ModMax Lagrangian (1.7) is not of the form $\sqrt{1+x}$ for some quantity $x$ involving field strengths (in fact it is non-analytic at $z = 0$), ModMax does not admit a derivative expansion of the same form. Of course, the Born-Infeld-like extension of ModMax [46], which we will define shortly, does possess an $\alpha^2$-type expansion, and this theory will be the main focus of our discussion.

There are other interesting properties of the ModMax theory that have recently been investigated — we will mention a few. Although the theory exhibits superluminal propagation when the deformation parameter $\gamma$ is negative, for $\gamma \geq 0$ it has well-behaved plane wave solutions. In particular, small-amplitude waves in the ModMax theory obey a polarization-dependent dispersion relation (birefringence), unlike the Born-Infeld theory. The ModMax theory has been shown to descend, via dimensional reduction, from a $6d$ theory of a chiral 2-form which can be described by a modified version of the Pasti-Sorokin-Tonin (PST) action [46]; for details on the original PST theory see [51–53]. Black hole solutions which are the analogues of the Reissner–Nordström black hole, but which are electrically charged under a gauge field described by ModMax, have been studied in [54–60].

Directly relevant for this paper are the Born-Infeld-like extensions of the $4d$ ModMax theory that have been constructed in [46] and then supersymmetrised in [47,48], see [61] for the $\mathcal{N} = 2$ case, obtaining an explicit example of the infinite class of supersymmetric duality invariant models defined in [39–44]. Written in terms of the $\gamma$ and $\alpha^2$ parameters, the Lagrangian for the Born-Infeld-ModMax theory takes the following form

$$\mathcal{L}_{\gamma\text{BI}} = \frac{1}{\alpha^2} \left\{ 1 - \sqrt{1 + \frac{\alpha^2}{2} \left[ \cosh(\gamma) F^2 - \sinh(\gamma) \sqrt{(F^2)^2 + (F\widetilde{F})^2} \right] - \frac{\alpha^4}{16}(F\widetilde{F})^2} \right\}. \qquad (1.8)$$

Considering the flow equations described in eq. (1.5b) for the Maxwell-Born-Infeld theory, together with its supersymmetric extension of [25], it is natural to wonder whether the whole one parameter family of Born-Infeld-like ModMax theories satisfies a $T^2$-like flow equation both in the non-supersymmetric and supersymmetric cases. The main purpose of this paper is in fact to analyse this query and to provide an affirmative answer to the following question: is the (supersymmetric) ModMax-BI Lagrangian satisfying a $T^2$-like flow for any $\gamma$? An intuition that this might be the case comes from the the auxiliary field formulation of duality invariant theories [38,62], see [44] for the supersymmetric case. In this framework, Maxwell theory with $\gamma = 0$ does not seem to have any special property compared to the ModMax case with $\gamma \neq 0$ [48,61]. This suggests that, if a Lagrangian flow exists for $\gamma = 0$ it should then exist for any $\gamma$, as we will indeed prove explicitly in our paper for the non-supersymmetric and $\mathcal{N} = 1$ supersymmetric cases.

This paper is organized as follows. In Section 2, we verify by direct computation that the Born-Infeld extension of the ModMax theory satisfies a $T^2$ flow for any value of the parameter $\gamma$. Section 3 then provides a different proof of this fact which begins from a general equation that applies to $T^2$ flows for Abelian gauge theories in any spacetime dimension. In Section 4, we extend this analysis to the case with $\mathcal{N} = 1$ supersymmetry, demonstrating that the supersymmetric extension of the ModMax-BI theory satisfies a supercurrent-squared flow equation which is the superspace analogue of the ordinary $T^2$ deformation. Section 5 summarizes these results and identifies some directions for future research. We also include two Appendices; in the first we elaborate on the equivalence of $T^2$ operators and $\sqrt{\det(T_{\mu\nu})}$ in $d = 4$, and in the

second we derive a general flow equation for $T^2$ flows in scalar theories for any spacetime dimension.

**Note Added:** During the preparation of this work the interesting paper [63] appeared with the overlapping result for the non-supersymmetric Born-Infeld like deformation of ModMax as a stress-tensor squared deformation. Interestingly, the authors of [63] also identified an operator which is a functional of the stress-energy tensor of ModMax-BI that triggers classically marginal (though non-analytic) deformations associated with the parameter $\gamma$. The supersymmetric extension of this result is an interesting venue for future research.

## 2 ModMax-BI is a $T^2$ flow

In [13] it was proven that the Maxwell-Born-Infeld theory with Lagrangian $\mathcal{L}_{\text{BI}}$,

$$
\begin{aligned}
S_{\text{BI}} = \int d^4x \, \mathcal{L}_{\text{BI}} &= \frac{1}{\alpha^2} \int d^4x \left[ 1 - \sqrt{-\det(\eta_{\mu\nu} + \alpha F_{\mu\nu})} \right] \\
&= \frac{1}{\alpha^2} \int d^4x \left[ 1 - \sqrt{1 + \frac{\alpha^2}{2} F^2 - \frac{\alpha^4}{16}(F\tilde{F})^2} \right] \\
&= -\frac{1}{4} \int d^4x \, F^2 + \text{higher derivative terms} ,
\end{aligned} \tag{2.1}
$$

satisfies the following flow equation with respect to the $\alpha^2$ parameter:

$$
\frac{\partial \mathcal{L}_{\text{BI}}}{\partial \alpha^2} = \frac{1}{8} O_{T^2} . \tag{2.2}
$$

The operator $O_{T^2}$ is defined as

$$
O_{T^2} \equiv T^{\mu\nu} T_{\mu\nu} - \frac{1}{2} \Theta^2, \qquad \Theta \equiv T^\mu_\mu , \tag{2.3}
$$

where $T^{\mu\nu}$ is the symmetric and conserved stress-energy tensor of the Maxwell-Born-Infeld theory. The previous composite operator is one of the representatives of an infinite family of stress-tensor squared operators of the following form:

$$
O_{T^2}^{[r]} = T^{\mu\nu} T_{\mu\nu} - r \, \Theta^2 . \tag{2.4}
$$

These are defined for any real constant parameter $r$ and stress-energy tensor of a relativistic QFT in $d$-dimensions. We will discuss some more examples in the next section. However, it is worth reminding that for $d = 2$ and $r = 1$, $O_{T^2}^{[1]}$ is the $T\bar{T}$ operator, which is proportional to $\det[T_{\mu\nu}]$, see [1–3]. In $d > 2$ it is still an open question whether there are operators that play the same role as $T\bar{T}$, and share the same remarkable properties. Notable proposals for extensions of $T\bar{T}$ in $d > 2$ are $O_{T^2}^{[1/(d-1)]}$ in $d$-dimensions, that were motivated from bulk cut-off holography [27, 28]. Rather than deforming a QFT by a flow triggered by generic $O_{T^2}^{[r]}$ (we will elaborate on general flows in the next section), in this section we are interested to check whether the Born-Infeld-like extension of the ModMax theory [32] satisfies a stress-tensor squared flow for a specific value of $r$. We will explicitly show that this is the case for $r = 1/2$, exactly as for the Born-Infeld Lagrangian (2.1).

The Born-Infeld-like extension of ModMax is defined by the following Lagrangian[2]

$$
\mathcal{L}_{\gamma\text{BI}} = t - \sqrt{t^2 - 2t \left[ \cosh(\gamma) S + \sinh(\gamma) \sqrt{S^2 + P^2} \right] - P^2} , \tag{2.5}
$$

---

[2]The parameter $t$ is the same as $T$ of [32].

where

$$S = -\frac{1}{4}F^{\mu\nu}F_{\mu\nu}, \quad P = -\frac{1}{4}F_{\mu\nu}\tilde{F}^{\mu\nu}, \quad \tilde{F}^{\mu\nu} = \frac{1}{2}\epsilon^{\mu\nu\lambda\tau}F_{\lambda\tau}, \tag{2.6}$$

and $F_{\mu\nu} = (\partial_\mu v_\nu - \partial_\nu v_\mu)$ is the field strength for an Abelian gauge field $v_\mu$. The $t \to +\infty$ limit leads to the ModMax Lagrangian

$$\mathcal{L}_{\mathrm{ModMax}} = \cosh(\gamma)S + \sinh(\gamma)\sqrt{S^2 + P^2}. \tag{2.7}$$

For $\gamma = 0$, and after identifying $t = 1/\alpha^2$, the Lagrangian $\mathcal{L}_{\gamma\mathrm{BI}}$ in eq. (2.5) turns into the Maxwell-Born-Infeld Lagrangian (2.1).

After minimally coupling the Born-Infeld-ModMax Lagrangian to a metric $g^{\mu\nu}$, it is a straightforward exercise to derive the Hilbert stress-energy tensor, given by[3]

$$T_{\mu\nu}^{\gamma\mathrm{BI}} = -\frac{2}{\sqrt{-g}}\frac{\delta S_{\gamma\mathrm{BI}}}{\delta g^{\mu\nu}}, \tag{2.8}$$

for (2.5). The result can be written as

$$T_{\mu\nu}^{\gamma\mathrm{BI}} = \eta_{\mu\nu}f_1(S,P) + f_2(S,P)F_\mu{}^\lambda F_{\nu\lambda}, \tag{2.9}$$

where the two functions $f_1(S,P)$ and $f_2(S,P)$ are defined as:

$$f_1(S,P) = \frac{t\left(-t + 2\cosh(\gamma)S + \sinh(\gamma)\frac{P^2+2S^2}{\sqrt{S^2+P^2}}\right)}{\sqrt{t^2 - 2t\left[\cosh(\gamma)S + \sinh(\gamma)\sqrt{S^2+P^2}\right] - P^2}} + t, \tag{2.10}$$

$$f_2(S,P) = \frac{t\left(\cosh(\gamma) + \sinh(\gamma)\frac{S}{\sqrt{S^2+P^2}}\right)}{\sqrt{t^2 - 2t\left[\cosh(\gamma)S + \sinh(\gamma)\sqrt{S^2+P^2}\right] - P^2}}. \tag{2.11}$$

The trace of the stress-energy tensor is

$$\Theta = \Theta(S,P) = \frac{4t\left(\cosh(\gamma)S + \sinh(\gamma)\sqrt{S^2+P^2} - t\right)}{\sqrt{t^2 - 2t\left[\cosh(\gamma)S + \sinh(\gamma)\sqrt{S^2+P^2}\right] - P^2}} + 4t. \tag{2.12}$$

It is worth underlining that the stress-energy tensor for the Born-Infeld-ModMax theory presented above is invariant under U(1) electro-magnetic duality transformations. This indicates that any deformation triggered by composite operators defined only in terms of the stress-energy tensor should remain electro-magnetic invariant.

Let us compute explicitly the $O_{T^2}$ operator, eq. (2.3). Thanks to the identity

$$(F\tilde{F})^2 = \frac{1}{4}(\epsilon_{\mu\nu\rho\sigma}F^{\mu\nu}F^{\rho\sigma})^2 = 4F_{\mu\nu}F^{\nu\rho}F_{\rho\sigma}F^{\sigma\mu} - 2(F^2)^2, \tag{2.13}$$

which implies

$$F_{\mu\nu}F^{\nu\rho}F_{\rho\sigma}F^{\sigma\mu} = 8S^2 + 4P^2, \tag{2.14}$$

the $O_{T^2}$ operator takes the form

$$O_{T^2} = 4f_1^2 + 4f_2^2\left(2S^2 + P^2\right) - 8f_1f_2S - \frac{1}{2}\Theta^2, \tag{2.15}$$

---

[3]In this subsection we work in a $d = 4$ Lorentzian space-time with mostly plus metric signature.

which, after plugging in the explicit expressions for $f_1(S,P)$, $f_2(S,P)$, and $\Theta(S,P)$, simplifies to

$$
\begin{aligned}
O_{T^2} = {} & \frac{8t^2}{(t-\mathcal{L}_{\gamma\text{BI}})^3} \Big\{ t^3 - 2P^2 t + t\cosh(2\gamma)(P^2+2S^2) + \cosh(\gamma)S(P^2-3t^2) \\
& + \sqrt{S^2+P^2}\big[2\sinh(2\gamma)St + \sinh(\gamma)(P^2-3t^2)\big]\Big\} \\
& + \frac{8t^2}{(t-\mathcal{L}_{\gamma\text{BI}})^2}\Big\{P^2-t^2+2t\sinh(\gamma)\sqrt{S^2+P^2}+2t\cosh(\gamma)S\Big\}\,,
\end{aligned}
\tag{2.16}
$$

where

$$
t-\mathcal{L}_{\gamma\text{BI}} = \sqrt{t^2 - 2t\big[\cosh(\gamma)S + \sinh(\gamma)\sqrt{S^2+P^2}\big] - P^2}\,.
\tag{2.17}
$$

Despite the seemingly involved expression, it is straightforward to directly check that (2.16) is the same as a derivative with respect to $t$, or equivalently with respect to $\alpha^2 = 1/t$, of the Born-Infeld-ModMax Lagrangian. More specifically, it holds

$$
\frac{\partial \mathcal{L}_{\gamma\text{BI}}}{\partial \alpha^2} = \frac{1}{8}O_{T^2} \qquad \Longleftrightarrow \qquad \frac{\partial \mathcal{L}_{\gamma\text{BI}}}{\partial t} = -\frac{1}{8t^2}O_{T^2}\,.
\tag{2.18}
$$

This remarkably shows that the stress-tensor squared operator leading the flow is the same for any value of $\gamma$.

# 3 Derivation from $T^2$ Master Flow Equation

We have seen in Section 2 that the Born-Infeld-like extension of the ModMax Lagrangian can be shown, via direct computation, to satisfy a $T\overline{T}$-like flow. In this section, we will present a complementary derivation of this result which begins from a general differential equation for $T^2$ flows involving an Abelian gauge theory in arbitrary spacetime dimension. We now briefly review this general flow equation, which first appeared in [30].

## 3.1 Review of General Flow Equation

For simplicity, in this section we will work in Euclidean signature.[4] In $d$ spacetime dimensions the field strength $F_{\mu\nu}$ associated with an Abelian gauge field $v_\mu$ can be thought of as a $d \times d$ matrix whose indices are raised or lowered with $\delta_{\mu\nu}$. By the Cayley-Hamilton theorem, every such matrix obeys its characteristic equation

$$
p(M) = M^d + c_{d-1}M^{d-1} + \cdots + c_1 M + (-1)^d \det(M)\mathbb{I}_d = 0\,,
\tag{3.1}
$$

where $\mathbb{I}_d$ is the $d \times d$ identity matrix. The constants $c_i$ are given by

$$
c_i = \sum_{\{k_l\}} \prod_{l=1}^{d} \frac{(-1)^{k_l+1}}{l^{k_l}k_l!}\big[\,\mathrm{tr}\big(M^l\big)\,\big]^{k_l}\,,
\tag{3.2}
$$

where the sum runs over all sets of non-negative integers $k_l$ which satisfy

$$
\sum_{l=1}^{d} l k_l = d - i\,.
\tag{3.3}
$$

---

[4]Our conventions follow those in Section 7.2.1 of [30], to which we refer the reader for more details. In particular, the choice of Euclidian signature in this section is made merely for convenience and does not substantively affect any of the results.

Because these $c_i$ are determined in terms of the lower traces $\operatorname{tr}(M^j)$ for $j = 1, \cdots, d$, equation (3.1) places a limit on the number of independent trace structures that a $d \times d$ matrix may have. In particular, given all of the traces

$$\operatorname{tr}(M), \operatorname{tr}(M^2), \cdots, \operatorname{tr}(M^d), \tag{3.4}$$

it follows that all higher traces $\operatorname{tr}(M^n)$ for $n > d$ can then be expressed in terms of the lower traces. We now restrict to the case of an antisymmetric matrix, appropriate for a field strength $F_{\mu\nu}$. The trace of any odd power of such a matrix vanishes, so a general scalar quantity built from $F_{\mu\nu}$ in $d$ dimensions can be expressed in terms of the independent traces $\operatorname{tr}(F^2)$, $\operatorname{tr}(F^4)$, $\cdots$, $\operatorname{tr}(F^{2k})$ where $k = \lfloor \frac{d}{2} \rfloor$. To ease notation, we define

$$x_i = \operatorname{tr}(F^{2i}), \tag{3.5}$$

for $i = 1, \cdots, k$.

Now consider a general Lagrangian for an Abelian gauge field with field strength $F_{\mu\nu}$ in $d$ Euclidean spacetime dimensions. Because the Lagrangian is a gauge-invariant scalar, it may therefore be written as a function of the $x_i$:[5]

$$\mathcal{L}(F) = \mathcal{L}(x_1, \cdots, x_k). \tag{3.6}$$

The Hilbert stress-energy tensor associated with this Lagrangian is

$$T_{\mu\nu} = \delta_{\mu\nu} \mathcal{L} - 2 \sum_{i=1}^{k} \frac{\partial \mathcal{L}}{\partial x_i} \cdot \frac{\delta x_i}{\delta g^{\mu\nu}} \bigg|_{g=\delta}. \tag{3.7}$$

Computing the metric derivative of one of the $x_j$ gives

$$\frac{\delta x_j}{\delta g^{\mu\nu}} = 2j F_{\mu\nu}^{2j}, \tag{3.8}$$

where we have introduced the notation

$$F_{\mu\nu}^{2j} = g^{\alpha_1 \beta_1} \cdots g^{\alpha_{2j-1} \beta_{2j-1}} F_{\mu\alpha_1} F_{\beta_1 \alpha_2} \cdots F_{\beta_{2j-2} \alpha_{2j-1}} F_{\beta_{2j-1} \nu}. \tag{3.9}$$

That is, $F_{\mu\nu}^{2j}$ is a product of $2j$ copies of $F_{\mu\nu}$ with all adjacent indices contracted except for the first and last. Using this in (3.7) yields a general expression for the stress-energy tensor,

$$T_{\mu\nu} = \delta_{\mu\nu} \mathcal{L} - 4 \sum_{i=1}^{k} i \frac{\partial \mathcal{L}}{\partial x_i} F_{\mu\nu}^{2i}. \tag{3.10}$$

The bilinears which appear in general $T^2$ flows are then

$$T^{\mu\nu} T_{\mu\nu} = \mathcal{L}^2 d - 8\mathcal{L} \sum_{i=1}^{k} i \frac{\partial \mathcal{L}}{\partial x_i} \operatorname{tr}(F^{2i}) + 16 \sum_{i,j=1}^{k} ij \frac{\partial \mathcal{L}}{\partial x_i} \frac{\partial \mathcal{L}}{\partial x_j} F^{2i,\mu\nu} F_{\mu\nu}^{2j},$$

$$\left(T_{\mu}^{\mu}\right)^2 = \mathcal{L}^2 d^2 - 8\mathcal{L} d \sum_{i=1}^{k} i \frac{\partial \mathcal{L}}{\partial x_i} \operatorname{tr}(F^{2i}) + 16 \sum_{i,j=1}^{k} ij \frac{\partial \mathcal{L}}{\partial x_i} \frac{\partial \mathcal{L}}{\partial x_j} \operatorname{tr}(F^{2i}) \operatorname{tr}(F^{2j}).$$

---

[5]In this paper we only consider manifestly gauge invariant Lagrangians of the form (3.6). In odd dimension it might be interesting to extend this ansatz and study flows involving terms that depend on both $F$ and Chern-Simons-like couplings. Note however that an ordinary Chern-Simons term is purely topological and would not contribute to the stress tensor.

A general flow by the operator $O_{T^2}^{[r]}$ defined in (2.4), described by the differential equation

$$\frac{\partial \mathcal{L}}{\partial \lambda} = T^{\mu\nu}T_{\mu\nu} - r\left(T^{\mu}{}_{\mu}\right)^2,$$

(3.11)

can therefore be written in terms of the $x_i$ as

$$\frac{\partial \mathcal{L}}{\partial \lambda} = (1-rd)d\mathcal{L}^2 - 8\mathcal{L}(1-rd)\sum_{i=1}^{k} i \frac{\partial \mathcal{L}}{\partial x_i} x_i + 16 \sum_{i,j=1}^{k} ij \frac{\partial \mathcal{L}}{\partial x_i}\frac{\partial \mathcal{L}}{\partial x_j}\left(x_{i+j} - r x_i x_j\right).$$

(3.12)

We will refer to (3.12) as the master flow equation, since it describes a general $T^2$ flow for a theory of a single Abelian field strength in $d$ dimensions. However, we note that (3.12) is not expressed in terms of the $k$ independent trace structures $x_1, \cdots, x_k$, since the quantity involving $x_{i+j}$ will introduce dependence on the higher traces. In applications of the master flow equations we must eliminate these variables in favor of the lower traces using the Cayley-Hamilton theorem, which produces dimension-dependent numerical factors.

We end this subsection by commenting on a condition regarding the stress-energy tensors for Abelian gauge theories, which is easy to understand using the formalism we have just reviewed. It was pointed out in [13] that the stress-energy tensor $T^{(\mathrm{BI})}$ for the Born-Infeld theory in four dimensions satisfies the condition

$$\sqrt{\det\left(T^{(\mathrm{BI})}\right)} = \frac{1}{4}\left(\frac{1}{2}\operatorname{tr}\left(T^{(\mathrm{BI})}\right)^2 - \operatorname{tr}\left(\left(T^{(\mathrm{BI})}\right)^2\right)\right).$$

(3.13)

Note that deformations driven by the determinant of the stress-energy tensor in higher dimensions appeared also in [11,64] were the operator $[\det(T)]^{1/(d-1)}$ was proposed as a $T\overline{T}$-like deformations in $d$ dimensions. At first sight, equation (3.13) seems like a fairly special constraint which might be related to the fact that the Born-Infeld Lagrangian satisfies a $T^2$ flow in $d = 4$. Indeed, the combination of stress-energy tensor bilinears appearing on the right side of (3.13) is proportional to $O_{T^2}$ and the determinant of the energy-momentum tensor is what defines the usual $T\overline{T}$ operator in two dimensions, so this constraint naively appears connected to this family of stress-energy tensor deformations.

However, the condition (3.13) in fact holds for the stress-energy tensor of *any* theory of an Abelian field strength in four spacetime dimensions (including, of course, the ModMax Lagrangian and its ModMax-BI generalization). The proof of this fact is a simple linear algebra exercise, which we have relegated to Appendix A, and relies only on the form of the Hilbert stress-energy tensor and the fact that the field strength $F_{\mu\nu}$ is antisymmetric. The upshot of this result is that we are free to think of our $T^2$ flow as being driven by the operator $O_{T^2}$ defined in (2.3), or by the operator $\sqrt{\det(T)}$, up to an overall constant scaling, when we are considering deformations of four-dimensional gauge theories.

## 3.2 Application to ModMax-BI Theory

We now specialize to the case of $d = 4$ spacetime dimensions, appropriate for the ModMax theory and its Born-Infeld-like extension. In this case, the two independent scalars that can be constructed from the field strength $F_{\mu\nu}$ are

$$x_1 = F_{\mu\nu}F^{\nu\mu} = \operatorname{tr}(F^2), \qquad x_2 = F^{\mu\sigma}F_{\sigma}{}^{\nu}F_{\nu}{}^{\rho}F_{\rho\mu} = \operatorname{tr}(F^4).$$

(3.14)

As we mentioned above, the master flow equation (3.12) will introduce dependence on the two higher traces $x_3 = \operatorname{tr}(F^6)$ and $x_4 = \operatorname{tr}(F^8)$. We must therefore eliminate these in terms of

$x_1$ and $x_2$. The constraint implied by the Cayley-Hamilton theorem (3.1) for a $4 \times 4$ matrix $M$ is

$$0 = M^4 - (\mathrm{tr}(M)) M^3 + \frac{1}{2} \left( (\mathrm{tr}(M))^2 - \mathrm{tr}(M^2) \right) M^2$$
$$- \frac{1}{6} \left( (\mathrm{tr}(M))^3 - 3 \, \mathrm{tr}(M^2) \, \mathrm{tr}(M) + 2 \, \mathrm{tr}(M^3) \right) M + \det(M) \mathbb{I}_4 . \tag{3.15}$$

We first take the trace of equation (3.15) and solve for the determinant to find

$$\det(M) = \frac{1}{24} \left( (\mathrm{tr}\, M)^4 - 6 \, \mathrm{tr}(M^2) (\mathrm{tr}\, M)^2 + 3 \left( \mathrm{tr}\, M^2 \right)^2 + 8 \, \mathrm{tr}(M) \, \mathrm{tr}(M^3) - 6 \, \mathrm{tr}(M^4) \right) . \tag{3.16}$$

Replacing $M$ with the antisymmetric matrix $F$, so that traces of odd powers vanish, gives

$$\det(F) = \frac{1}{8} \left( \mathrm{tr}\left( F^2 \right) \right)^2 - \frac{1}{4} \, \mathrm{tr}(F^4) . \tag{3.17}$$

If we had first multiplied equation (3.15) by $M^2$ or by $M^4$ before taking the trace, we would have obtained the conditions

$$\mathrm{tr}(F^6) = \frac{1}{2} \, \mathrm{tr}(F^2) \, \mathrm{tr}(F^4) - \det(F) \, \mathrm{tr}(F^2) , \tag{3.18a}$$

$$\mathrm{tr}(F^8) = \frac{1}{2} \, \mathrm{tr}(F^2) \, \mathrm{tr}(F^6) - \det(F) \, \mathrm{tr}(F^4) . \tag{3.18b}$$

This system of equations can be solved and written in terms of the variables $x_i = \mathrm{tr}(F^{2i})$, which yields

$$x_3 = -\frac{1}{8} x_1 \left( x_1^2 - 6 x_2 \right) , \qquad x_4 = -\frac{1}{16} \left( x_1^4 - 4 x_1^2 x_2 - 4 x_2^2 \right) . \tag{3.19}$$

Substituting the expressions (3.19) into the master flow equation (3.12) and setting $d = 4$, we obtain the general differential equation

$$\frac{\partial \mathcal{L}}{\partial \lambda} = (4 - 16 r) \mathcal{L}^2 - 8 (1 - 4r) \mathcal{L} \left( x_1 \frac{\partial \mathcal{L}}{\partial x_1} + 2 x_2 \frac{\partial \mathcal{L}}{\partial x_2} \right) + 16 \left( \frac{\partial \mathcal{L}}{\partial x_1} \right)^2 \left( x_2 - r x_1^2 \right)$$
$$+ 16 \left( -\frac{1}{2} \frac{\partial \mathcal{L}}{\partial x_1} \frac{\partial \mathcal{L}}{\partial x_2} x_1 \left( x_1^2 - 6 x_2 + 8 r x_2 \right) + \left( \frac{\partial \mathcal{L}}{\partial x_2} \right)^2 \left( -\frac{1}{4} x_1^4 + x_1^2 x_2 + (1 - 4r) x_2^2 \right) \right) . \tag{3.20}$$

We now wish to show that equation (3.20) admits a solution which reduces to the ModMax theory when $\lambda = 0$, and that this solution exists only for the value $r = \frac{1}{2}$ of the relative coefficient in the deformation. We do this by first making a slightly more general ansatz and then demonstrating that consistency with the flow equation requires the ansatz to take exactly the form of the ModMax-BI Lagrangian.

More precisely, we begin by making an ansatz of the form[6]

$$\mathcal{L}(\lambda) = \frac{1}{\alpha \lambda} \left( \sqrt{1 + 2 \alpha \lambda \left( e^{-\gamma} u(x_1, x_2) + e^{\gamma} v(x_1, x_2) \right) + 4 \alpha^2 \lambda^2 u(x_1, x_2) v(x_1, x_2)} - 1 \right) , \tag{3.21}$$

and we further assume that the functions $u, v$ can be written as a sum and difference as

$$u(x_1, x_2) = \beta x_1 + w(x_1, x_2) , \qquad v(x_1, x_2) = \beta x_1 - w(x_1, x_2) . \tag{3.22}$$

---

[6]Do not confuse the numerical constant $\alpha$ used in this section with the dimensionful coupling constant $\alpha^2$ in sections 1, 2 and 4 of the paper, as, for example, in equations (1.5a) and (1.8).

At this stage, $w(x_1, x_2)$ is an arbitrary function of the two scalars that can be constructed from $F_{\mu\nu}$, while $\alpha, \beta$ are undetermined numerical constants. We now obtain constraints on these quantities from consistency with the master flow equation in $d = 4$. From demanding that the ansatz (3.21) be consistent with the differential equation (3.20) at zeroth order in both $\lambda$ and $\gamma$, one finds that the function $w$ must be

$$w(x_1, x_2) = \frac{2\sqrt{2}\beta}{\sqrt{\alpha}}\sqrt{x_1^2 - 4x_2}\,. \tag{3.23}$$

Substituting this expression for $w$ into the flow equation and then expanding to first order in $\lambda$ but zeroth order in $\gamma$ then produces the constraint

$$\alpha = -8\,. \tag{3.24}$$

Finally, we expand the flow equation to second order in $\lambda$ and to first order in $\gamma$, after using the above results for $w$ and $\alpha$, and find that the differential equation at this order is satisfied only if

$$r = \frac{1}{2}\,. \tag{3.25}$$

After imposing these various conditions, we arrive at

$$\mathcal{L}(\lambda) = \frac{1}{8\lambda}\left(1 - \sqrt{1 - 32\beta\lambda\left(x_1\cosh(\gamma) + \sinh(\gamma)\sqrt{4x_2 - x_1^2}\right) + 512\beta^2\lambda^2(x_1^2 - 2x_2)}\right). \tag{3.26}$$

This Lagrangian is an exact solution to the $4d$ master flow equation (3.20) to all orders in $\lambda$ and $\gamma$, and with any choice of the arbitrary constant $\beta$. However, to make contact with the preceding section, it is convenient to make a few changes of conventions. First, since (3.26) is a solution for any choice of the normalization $\beta$, we are free to choose $\beta = \frac{1}{32}$. Further, we can eliminate the variables $x_1 = \mathrm{tr}(F^2)$ and $x_2 = \mathrm{tr}(F^4)$ in terms of the variables $S, P$ defined in equation (2.6). The dictionary which translates between these variables is

$$x_1 = 4S\,, \qquad x_2 = 4P^2 + 8S^2\,. \tag{3.27}$$

After making these replacements, we find

$$\mathcal{L}(\lambda) = \frac{1}{8\lambda}\left(1 - \sqrt{1 - 4\lambda\left(\cosh(\gamma)S + \sinh(\gamma)\sqrt{S^2 + P^2}\right) - 4P^2\lambda^2}\right). \tag{3.28}$$

This is exactly the form of the Born-Infeld extension of the ModMax Lagrangian which we first defined in equation (2.5) after making the identification $t = \frac{1}{8\lambda}$.

## 3.3 No ModMax-BI Solutions to $T^2$ Flows in $d > 4$

The properties of the ModMax Lagrangian (1.7), and its Born-Infeld extension, are special to four dimensions because they are written in terms of the dual field strength $\tilde{F}_{\mu\nu}$, and the Hodge dual of $F_{\mu\nu}$ would be a higher $p$-form in $d > 4$ spacetime dimensions. Thus a ModMax-like theory in $d > 4$ would, of course, not exhibit any analogue of duality invariance. However, as a pure statement about $T^2$ flows, one could ask whether the square-root structure appearing in the ModMax-BI theory is a generic feature of deformations by stress-energy tensor bilinears, or whether it is also special to flows in $d = 4$. We saw in equation (3.26) that the ModMax-BI Lagrangian can be written in the form

$$\mathcal{L}(\lambda) = \frac{1}{8\lambda}\left(1 - \sqrt{1 - \lambda\left(x_1\cosh(\gamma) + \sinh(\gamma)\sqrt{4x_2 - x_1^2}\right) + \frac{1}{2}\lambda^2(x_1^2 - 2x_2)}\right), \tag{3.29}$$

where for convenience we repeat the definitions of the two indpendent scalars $x_1, x_2$ that can be constructed from a field strength in four dimensions:

$$x_1 = F_{\mu\nu} F^{\nu\mu} = \text{tr}(F^2), \qquad x_2 = F^{\mu\sigma} F_\sigma{}^\nu F_\nu{}^\rho F_{\rho\mu} = \text{tr}(F^4). \qquad (3.30)$$

Although (3.29) is equivalent to the ModMax-BI Lagrangian, it is not written in terms of $\widetilde{F}_{\mu\nu}$ and therefore makes sense in any number of spacetime dimensions $d \geq 4$ (the cases for $d < 4$ are trivial because $x_1$ and $x_2$ are no longer independent). Thus one might ask whether a Lagrangian of the form (3.29) satisfies a $T^2$ flow in any higher spacetime dimension.[7]

We will now show that the answer to this question is no. Suppose that we make an ansatz for a $d$-dimensional Lagrangian which is inspired by the ModMax-BI theory and thus only depends on the first two traces $x_1, x_2$:

$$\mathcal{L}(\lambda) = \frac{1}{\alpha\lambda} \left( \sqrt{1 + 2\alpha\lambda \left( e^{-\gamma} u(x_1, x_2) + e^\gamma v(x_1, x_2) \right) + 4\alpha^2 \lambda^2 u(x_1, x_2) v(x_1, x_2)} - 1 \right). \qquad (3.31)$$

Furthermore, we would like our ansatz to reduce to the free Maxwell action when we take both $\gamma = 0$ and $\lambda = 0$, so we should make the same refinement to our ansatz as in (3.22) for the $d = 4$ case:

$$u(x_1, x_2) = \beta x_1 + w(x_1, x_2), \qquad v(x_1, x_2) = \beta x_1 - w(x_1, x_2). \qquad (3.32)$$

When $\lambda = 0$, this reduces to an undeformed theory of the form

$$\mathcal{L}(0) = e^\gamma (\beta x_1 - w) + e^{-\gamma} (\beta x_1 + w). \qquad (3.33)$$

On dimensional grounds, the function $w$ must be proportional either to $x_1$, or to $\sqrt{x_2}$, or to a general combination $\sqrt{c_1 x_1^2 + c_2 x_2}$ which has the same dimension as $x_1$. Therefore we will assume that $w$ can be written as

$$w(x_1, x_2) = \sqrt{c_1 x_1^2 + c_2 x_2}, \qquad (3.34)$$

which is the same form as in the usual $d = 4$ ModMax-BI Lagrangian.

We must check whether any choice of the constants $\alpha, \beta, c_1, c_2$ makes this ansatz consistent with the master flow equation (3.12), which we repeat:

$$\frac{\partial \mathcal{L}}{\partial \lambda} = (1 - rd) d\mathcal{L}^2 - 8\mathcal{L}(1 - rd) \sum_{i=1}^k i \frac{\partial \mathcal{L}}{\partial x_i} x_i + 16 \sum_{i,j=1}^k ij \frac{\partial \mathcal{L}}{\partial x_i} \frac{\partial \mathcal{L}}{\partial x_j} \left( x_{i+j} - r x_i x_j \right). \qquad (3.35)$$

There are two cases to consider.

1. $d \geq 6$. In this case, there is at least one additional independent trace structure $x_3$. Because the ansatz (3.31) is a function only of $x_1$ and $x_2$, the left side of the master flow equation is independent of $x_3$. However, the right side of the flow equation contains a term $\frac{\partial \mathcal{L}}{\partial x_1} \frac{\partial \mathcal{L}}{\partial x_2} x_3$ which is non-zero and depends on $x_3$. There is no constraint relating $x_3$ to other traces in $d \geq 6$, so the two sides cannot be equal and therefore the ansatz does not solve the flow equation.

---

[7]The analogous question of whether the ordinary Born-Infeld Lagrangian arises from a $T^2$ flow in $d > 4$ was answered in the negative in [30]. However, the naive generalization (3.31) of ModMax-BI does not reduce to Born-Infeld when $\gamma = 0$ except in $d = 4$. Therefore the absence of Born-Infeld solutions to $T^2$ flows in $d > 4$ does not imply anything about the absence of ModMax-BI solutions.

2. $d = 5$. In this case, $x_3$ is not independent of $x_1$ and $x_2$, but rather satisfies

$$x_3 = \frac{3}{4} x_1 x_2 - \frac{1}{8} x_1^3, \tag{3.36}$$

by the Cayley-Hamilton theorem. Similarly,

$$x_4 = \frac{1}{16} \left( 4 x_2^2 + 4 x_1^2 x_2 - x_1^4 \right). \tag{3.37}$$

One can substitute these relations into the master flow equation and then impose consistency order-by-order in $\lambda$. The constraint which is first order in $\lambda$ will be satisfied so long as

$$\alpha = 8, \quad r = 1, \quad c_1 = -\beta^2, \quad c_2 = 4\beta^2. \tag{3.38}$$

However, upon expanding to second order in $\lambda$, one finds that the ansatz cannot be made consistent with the flow equation for any non-zero choice of the remaining parameter $\beta$.

Therefore we see that the naive generalization (3.31) of the ModMax-BI theory only satisfies a $T^2$ flow in $d = 4$, similar to the Born-Infeld action.

Rather than the question that we have addressed above, one could also ask a slightly more general question which exploits the fact that in $d > 4$ there are more independent scalars than $x_1$ and $x_2$. Thus one might wonder whether a different Lagrangian, which still possesses a square root of the form appearing in (3.29) but whose argument depends on $x_1, x_2, x_3$ and higher trace structures, satisfies a $T^2$ flow in higher dimension.

For instance, one could make another ansatz of the form

$$\mathcal{L}(\lambda) = \frac{1}{\alpha\lambda} \left( \sqrt{1 + 2\alpha\lambda \left( e^{-\gamma} u(x_i) + e^{\gamma} v(x_i) \right) + 4\alpha^2 \lambda^2 u(x_i) v(x_i)} - 1 \right), \tag{3.39}$$

where the functions $u(x_i)$, $v(x_i)$ now depend on all trace structures $x_1, \cdots, x_k$. We still assume that

$$u(x_1, \cdots, x_k) = \beta x_1 + w(x_1, \cdots, x_k), \qquad v(x_1, \cdots, x_k) = \beta x_1 - w(x_1, \cdots, x_k), \tag{3.40}$$

but now allow a more general form of the function $w$, such as

$$w(x_i) = \sqrt[k]{c_1 x_1^k + c_2 x_2 x_1^{k-1} + \cdots + c_N x_k}, \tag{3.41}$$

which reduces to our previous ansatz when $k = 2$ and which is again required on dimensional grounds since $w$ must have the same dimension as $x_1$. For instance, one could consider a six-dimensional analogue of the ModMax-BI theory where

$$w(x_1, x_2, x_3) = \sqrt[3]{c_1 x_1^3 + c_2 x_1 x_2 + c_3 x_3}. \tag{3.42}$$

As an example, we will explicitly check whether the six-dimensional ansatz using (3.42) can solve the master flow equation for any choice of parameters. This will require one additional use of the Cayley-Hamilton theorem, with coefficients appropriate for $6 \times 6$ matrices, in order to eliminate higher traces in the master flow equation. In this case, the equation obeyed by the field strength $F$ is

$$0 = F^6 - \frac{1}{2} x_1 F^4 + \frac{1}{8} \left( x_1^2 - 2 x_2 \right) F^2 + \det(M) \mathbb{I}_6 = 0. \tag{3.43}$$

By repeatedly using (3.43) in the same way as above, we can express various higher traces in terms of the three independent structures $x_1, x_2, x_3$. In particular,

$$
\begin{aligned}
x_4 &= \frac{1}{48}\left(x_1^4 - 12x_1^2 x_2 + 12x_2^2 + 32x_1 x_3\right), \\
x_5 &= \frac{1}{96}\left(x_1^5 - 10x_1^3 x_2 + 20x_1^2 x_3 + 40x_2 x_3\right), \\
x_6 &= \frac{1}{384}\left(x_1^6 - 6x_1^4 x_2 - 36x_1^2 x_2^2 + 24x_2^3 + 16x_1^3 x_3 + 96x_1 x_2 x_3 + 64x_3^2\right).
\end{aligned}
\tag{3.44}
$$

We may now substitute these trace expressions into the master flow equation (3.12) and set $d = 6$. The resulting differential equation is rather unwieldy, but we record it here for completeness:

$$
\begin{aligned}
\frac{\partial \mathcal{L}}{\partial \lambda} ={}& 6\mathcal{L}^2(1-6r) + 16\left(\frac{\partial \mathcal{L}}{\partial x_1}\right)^2\left(x_2 - rx_1^2\right) + 8\mathcal{L}(6r-1)\left(\frac{\partial \mathcal{L}}{\partial x_1}x_1 + 2\frac{\partial \mathcal{L}}{\partial x_2}x_2 + 3\frac{\partial \mathcal{L}}{\partial x_3}x_3\right) \\
&+ \frac{4}{3}\left(\frac{\partial \mathcal{L}}{\partial x_2}\right)^2\left(x_1^4 - 12x_1^2 x_2 + 12(1-4r)x_2^2 + 32x_1 x_3\right) \\
&+ \frac{4}{3}\frac{\partial \mathcal{L}}{\partial x_2}\frac{\partial \mathcal{L}}{\partial x_3}\left(x_1^5 - 10x_1^3 x_2 + 20x_1^2 x_3 + 8(5-12r)x_2 x_3\right) \\
&+ \frac{3}{8}\left(\frac{\partial \mathcal{L}}{\partial x_3}\right)^2\left(x_1^6 - 6x_1^4 x_2 - 36x_1^2 x_2^2 + 24x_2^3 + 16x_1(x_1^2 + 6x_2)x_3 + 64(1-6r)x_3^2\right) \\
&+ 2\frac{\partial \mathcal{L}}{\partial x_1}\left(32\frac{\partial \mathcal{L}}{\partial x_2}(x_3 - rx_1 x_2) + \frac{\partial \mathcal{L}}{\partial x_3}\left(x_1^4 - 12x_1^2 x_2 + 12x_2^2 + 16(2-3r)x_1 x_3\right)\right).
\end{aligned}
\tag{3.45}
$$

Upon substituting the ansatz involving the expression $w$ in (3.42) into the $6d$ master flow equation (3.45), one finds that no choice of the parameters is consistent with the differential equation even at the lowest order in $\lambda$ and $\gamma$. One way to see this is to consider the $\gamma = 0$ limit and note that $\mathcal{L}(\lambda = 0, \gamma = 0)$ is proportional to the Maxwell Lagrangian $x_1 = \text{tr}(F^2)$. Therefore, the leading deformation from $T\overline{T}$ will only introduce terms involving $x_2$ and $x_1^2$. But the $\mathcal{O}(\lambda)$ expansion of (3.39) also includes dependence on $x_3$ (if $c_3 \neq 0$), which cannot be consistent since $x_3$ is independent from $x_1$ and $x_2$. Therefore there is no Lagrangian of this form which satisfies a $T^2$ flow in six dimensions. A similar argument can be used to show that no other ansatz involving a function $w(x_i)$ as in (3.41) can be consistent with a $T\overline{T}$ flow in any higher number of spacetime dimensions.

This concludes the proof that no obvious analogue of the ModMax-BI theory satisfies a $T\overline{T}$ flow in any dimension other than $d = 4$. It is possible that such a generalized ModMax-BI Lagrangian might obey a flow equation driven by some other combination of stress tensors, such as an operator of the form $(\det T)^p$ for some power $p$ or a scalar built from contractions of three or more copies of $T_{\mu\nu}$. We will not undertake a general analysis of deformations driven by other stress tensor combinations here. However, if it is true that some ModMax-BI-like theory can be viewed as a deformation by some special stress tensor operator other than $O_{T^2}^{[r]}$, one could use this as a principle which identifies a preferred higher-dimensional analogue of the ModMax-BI action.

## 4 Supersymmetric ModMax-BI is a supercurrent-squared flow

Let us now turn to the $d = 4$, $\mathcal{N} = 1$ supersymmetric case. We first review the result of [25] concerning the extension of the flow eq. (2.2) for the supersymmetric Maxwell-Born-Infeld theory introducing the associated supercurrent-squared operator. We will then discuss the

extension to the Born-Infeld-like ModMax theory of [47] and show that the same operator introduced in [25] drives a flow equation with respect to the $\alpha^2$ parameter.

## 4.1 Review of the Bagger-Galperin Lagrangian flow and the $d = 4$, $\mathcal{N} = 1$ super-current squared operator

The supersymmetric extension of the Maxwell-Born-Infeld Lagrangian proposed by Bagger and Galperin in [31] is defined by the following $d = 4$, $\mathcal{N} = 1$ Lagrangian in superspace:

$$\mathcal{L}_{\text{susy-BI}} = \frac{1}{4}\left[ \int d^2\theta \, W^2 + \int d^2\bar{\theta} \, \bar{W}^2 + \int d^2\theta d^2\bar{\theta} \, \frac{\alpha^2 W^2 \bar{W}^2}{1 - \alpha^2 \mathbb{S} + \sqrt{1 - 2\alpha^2 \mathbb{S} - \alpha^4 \mathbb{P}}} \right]. \quad (4.1)$$

Here the superfields $\mathbb{S}$ and $\mathbb{P}$ are[8]

$$\mathbb{S} = -\frac{1}{16}(D^2 W^2 + \bar{D}^2 \bar{W}^2), \quad \mathbb{P} = \frac{i}{16}(D^2 W^2 - \bar{D}^2 \bar{W}^2), \quad (4.2)$$

with $W_\alpha$, and its conjugate $\bar{W}_{\dot{\alpha}} = (W_\alpha)^*$, being the superfield strength of a $d = 4$, $\mathcal{N} = 1$ Abelian vector multiplet obeying:

$$\bar{D}_{\dot{\beta}} W_\alpha = 0, \quad D^\alpha W_\alpha = \bar{D}_{\dot{\alpha}} \bar{W}^{\dot{\alpha}}. \quad (4.3)$$

In components, $W_\alpha$ has the following expansion in terms of the fields describing the vector multiplet

$$W_\alpha = -i\lambda_\alpha + \theta_\alpha D - i(\sigma^{\mu\nu}\theta)_\alpha F_{\mu\nu} + \theta^2(\sigma^\mu \partial_\mu \bar{\lambda})_\alpha, \quad (4.4)$$

where the complex spinor $\lambda_\alpha$ is the gaugino, $D$ is the real auxiliary field, and $F_{\mu\nu} = 2\partial_{[\mu} v_{\nu]}$ is the field strength of an Abelian connection $v_\mu$. The superfields $\mathbb{S}$ and $\mathbb{P}$ are such that their $\theta = 0$ components give $S$ and $P$ of eq. (2.6)

$$\mathbb{S}|_{\theta=0} = S + \frac{1}{2}D^2, \quad \mathbb{P}|_{\theta=0} = P, \quad (4.5)$$

up to a $D = D^\alpha W_\alpha|_{\theta=0} = \bar{D}_{\dot{\alpha}} \bar{W}^{\dot{\alpha}}|_{\theta=0}$ term.

In [25] it was shown that the Bagger-Galperin supersymmetric extension of the Maxwell-Born-Infeld theory satisfies a flow equation driven by a supercurrent-squared operator. More precisely, up to an on-shell condition, that we will review and use also in the general ModMax case, the Lagrangian (4.1) was shown to satisfy[9]

$$\frac{\partial \mathcal{L}_{\text{susy-BI}}}{\partial \alpha^2} = \frac{1}{8} \int d^2\theta d^2\bar{\theta} \, \mathcal{O}_{T^2}, \quad (4.6)$$

where $\mathcal{O}_{T^2}$ is

$$\mathcal{O}_{T^2} = \frac{1}{16} \mathcal{J}^{\alpha\dot{\alpha}} \mathcal{J}_{\alpha\dot{\alpha}} - \frac{5}{8} \mathcal{X}\bar{\mathcal{X}}. \quad (4.7)$$

The operator $\mathcal{O}_{T^2}$ is defined in terms of the superfields of the Ferrara-Zumino (FZ) supercurrent multiplet [66]. The explicit form of $\mathcal{J}_{\alpha\dot{\alpha}}$, $\mathcal{X}$, and $\mathcal{O}_{T^2}$ for the Bagger-Galperin model (4.1) were computed in [25] by using results of [41]. We will extend this analysis in the next subsection.

In general, the vector superfield $\mathcal{J}_{\alpha\dot{\alpha}}$ and the complex scalar superfield $\mathcal{X}$ of the FZ multiplet satisfy the following constraints:

$$\bar{D}^{\dot{\alpha}} \mathcal{J}_{\alpha\dot{\alpha}} = D_\alpha \mathcal{X}, \quad \bar{D}_{\dot{\alpha}} \mathcal{X} = 0. \quad (4.8)$$

---

[8]We use the notation $D^2 := D^\alpha D_\alpha$, $W^2 := W^\alpha W_\alpha$, $\bar{D}^2 := \bar{D}_{\dot{\alpha}} \bar{D}^{\dot{\alpha}}$, and $\bar{W}^2 := \bar{W}_{\dot{\alpha}} \bar{W}^{\dot{\alpha}}$. For more detail concerning our notation we refer the reader to [25].

[9]In [65] Cecotti and Ferrara were the first to observe the flow at order $\alpha^2$ where $\mathcal{O}_{T^2} = W^2 \bar{W}^2 + \cdots$.

These constraints lead to $12 + 12$ independent component fields that appear in $\mathcal{J}_{\alpha\dot\alpha}$ and $\mathcal{X}$ as follows[10]

$$
\begin{aligned}
\mathcal{J}_\mu(x,\theta,\bar\theta) \;=\;& j_\mu + \theta\Big(S_\mu - \frac{1}{\sqrt{2}}\sigma_\mu\bar\chi\Big) + \bar\theta\Big(\bar S_\mu + \frac{1}{\sqrt{2}}\bar\sigma_\mu\chi\Big) + \frac{i}{2}\theta^2\partial_\mu\bar{\mathsf{x}} - \frac{i}{2}\bar\theta^2\partial_\mu\mathsf{x} \\
&+ \theta\sigma^\nu\bar\theta\Big(2T_{\mu\nu} - \frac{2}{3}\eta_{\mu\nu}\Theta - \frac{1}{2}\epsilon_{\nu\mu\rho\sigma}\partial^\rho j^\sigma\Big) - \frac{i}{2}\theta^2\bar\theta\Big(\bar{\slashed\partial}S_\mu + \frac{1}{\sqrt{2}}\bar\sigma_\mu\slashed\partial\bar\chi\Big) \\
&- \frac{i}{2}\bar\theta^2\theta\Big(\slashed\partial\bar S_\mu - \frac{1}{\sqrt{2}}\sigma_\mu\bar{\slashed\partial}\chi\Big) + \frac{1}{2}\theta^2\bar\theta^2\Big(\partial_\mu\partial^\nu j_\nu - \frac{1}{2}\partial^2 j_\mu\Big),
\end{aligned}
\tag{4.9a}
$$

$$
\mathcal{X}(y,\theta) \;=\; \mathsf{x}(y) + \sqrt{2}\theta\chi(y) + \theta^2\mathsf{F}(y), \quad \chi_\alpha = \frac{\sqrt{2}}{3}(\sigma^\mu)_{\alpha\dot\alpha}\bar S_\mu^{\dot\alpha}, \quad \mathsf{F} = \frac{2}{3}\Theta + i\partial_\mu j^\mu. \tag{4.9b}
$$

The operators $T_{\mu\nu}$ and $\Theta$ are the conserved stress-energy tensor and its trace, while $(S_\mu{}^\alpha, \bar S_{\mu\dot\alpha})$ is the conserved $d = 4, \mathcal{N} = 1$ supersymmetry current. The other operators in the FZ multiplet are required by supersymmetry. Note that, in general, $j_\mu$ is not a conserved vector.

The flow equation (4.6) for the Bagger-Galperin theory leads to a definition of a manifestly supersymmetric extension of the operator (2.3) and an associated flow for any $d = 4, \mathcal{N} = 1$ supersymmetric QFT admitting a Ferrara-Zumino supercurrent multiplet [25]. In fact, up to total derivatives and the (on-shell) conservation equations (4.8), the following result holds:

$$
\begin{aligned}
\int d^4\theta\, \mathcal{O}_{T^2} \;=\;& \Big(T^{\mu\nu}T_{\mu\nu} - \frac{1}{2}\Theta^2\Big) + \frac{3}{8}j_\mu\partial^2 j^\mu + \frac{3}{8}\partial_\mu\mathsf{x}\partial^\mu\bar{\mathsf{x}} - \frac{i}{2}\Big(S_\mu\slashed\partial\bar S^\mu - \frac{9}{4}\bar\chi\bar{\slashed\partial}\chi\Big) \\
&+ \text{total derivatives} + \text{EOM}.
\end{aligned}
\tag{4.10}
$$

The first two terms in the first bracket are precisely the operator $\mathcal{O}_{T^2}$ of eq. (2.3), while the extra terms in $\mathcal{O}_{T^2}$ are required by $4d, \mathcal{N} = 1$ supersymmetry.

## 4.2 The Born-Infeld-ModMax case

The Lagrangian density for a Born-Infeld-like extension of the supersymmetric ModMax theory was derived in [47]. It takes the following form

$$
\mathcal{L}_{\text{susy}-\gamma\text{BI}} \;=\; \frac{\cosh(\gamma)}{4}\left\{\int d^2\theta\, W^2 + \int d^2\bar\theta\, \bar W^2 + \int d^2\theta d^2\bar\theta\, W^2\bar W^2 K(\mathbb{S},\mathbb{P})\right\}, \tag{4.11a}
$$

with the superfield $K(\mathbb{S},\mathbb{P})$ given by

$$
K(\mathbb{S},\mathbb{P}) \;=\; \frac{t - \sqrt{t^2 - 2t\big[\cosh(\gamma)\mathbb{S} + \sinh(\gamma)\sqrt{\mathbb{S}^2 + \mathbb{P}^2}\big] - \mathbb{P}^2} - \cosh(\gamma)\mathbb{S}}{\cosh(\gamma)(\mathbb{S}^2 + \mathbb{P}^2)}. \tag{4.11b}
$$

In the limit of $\gamma = 0$ the Lagrangian (4.11) reduces to the Bagger-Galperin Lagrangian (4.1) upon identifying $t = 1/\alpha^2$. The aim of this subsection is to prove that $\mathcal{L}_{\text{susy}-\gamma\text{BI}}$ satisfies the flow equation

$$
\frac{\partial\mathcal{L}_{\text{susy}-\text{BI}}}{\partial\alpha^2} = \frac{1}{8}\int d^2\theta d^2\bar\theta\, \mathcal{O}_{T^2} \quad\Longleftrightarrow\quad \frac{\partial\mathcal{L}_{\text{susy}-\text{BI}}}{\partial t} = -\frac{1}{8t^2}\int d^2\theta d^2\bar\theta\, \mathcal{O}_{T^2}. \tag{4.12}
$$

A first step towards proving such a flow equation is to compute the Ferrara-Zumino multiplet, and then the operator $\mathcal{O}_{T^2}$ of eq. (4.7), for the $\mathcal{L}_{\text{susy}-\text{BI}}$ Lagrangian. It is useful to rewrite the Lagrangian (4.11), and in particular $K(\mathbb{S},\mathbb{P})$, in terms of the following superfields:

$$
u = \frac{1}{8}D^2 W^2, \qquad \bar u = \frac{1}{8}\bar D^2 \bar W^2, \tag{4.13}
$$

---

[10]The complex chiral coordinate $y^\mu$ is defined as $y^\mu = x^\mu + i\theta\sigma^\mu\bar\theta$, while the slashed derivatives are $\slashed\partial = \sigma^\mu\partial_\mu$, $\bar{\slashed\partial} = \bar\sigma^\mu\partial_\mu$.

such that

$$\mathbb{S} = -\frac{1}{2}(u+\bar{u}), \qquad \mathbb{P} = \frac{i}{2}(u-\bar{u}), \qquad \mathbb{S}^2 + \mathbb{P}^2 = u\bar{u}. \qquad (4.14)$$

The superfield $K$ then satisfies

$$K(u,\bar{u}) = \frac{u+\bar{u}-\text{sech}(\gamma)\left[\sqrt{4t^2-8t\sinh(\gamma)\sqrt{u\bar{u}}+4t\cosh(\gamma)(u+\bar{u})+(u-\bar{u})^2}-2t\right]}{2u\bar{u}}. \qquad (4.15)$$

The Ferrara-Zumino supercurrent for a large class of models of the form

$$\mathcal{L}_\Lambda = \frac{1}{4}\int d^2\theta\, W^2 + \frac{1}{4}\int d^2\bar{\theta}\, \bar{W}^2 + \frac{1}{4}\int d^2\theta d^2\bar{\theta}\, W^2\bar{W}^2\Lambda(u,\bar{u}), \qquad (4.16)$$

was computed in [41] by using superspace techniques for $d=4$, $\mathcal{N}=1$ old-minimal Poincaré supergravity.[11] We can readily use the results of [41] to write the expressions that we need for the FZ multiplet derived from $\mathcal{L}_{\text{susy}-\text{BI}}$. These are

$$\mathcal{X} = \frac{\cosh(\gamma)}{6} W^2\bar{D}^2\big(\bar{W}^2(\Gamma+\bar{\Gamma}-K)\big), \qquad (4.17a)$$

$$\mathcal{J}_{\alpha\dot{\alpha}} = \cosh(\gamma)\left\{-2iM_\alpha\bar{W}_{\dot{\alpha}} + 2iW_\alpha\bar{M}_{\dot{\alpha}} + \frac{1}{12}[D_\alpha,\bar{D}_{\dot{\alpha}}](W^2\bar{W}^2)\cdot\big(\Gamma+\bar{\Gamma}-K\big)\right\}$$
$$+ W^2\bar{W}(\cdots) + \bar{W}^2 W(\cdots). \qquad (4.17b)$$

Here $\Gamma = \Gamma(u,\bar{u})$ and $\bar{\Gamma} = \bar{\Gamma}(u,\bar{u})$ are

$$\Gamma(u,\bar{u}) = \frac{\partial(uK(u,\bar{u}))}{\partial u}, \qquad \bar{\Gamma}(u,\bar{u}) = \frac{\partial(\bar{u}K(u,\bar{u}))}{\partial\bar{u}}, \qquad (4.18)$$

while the superfield $M_\alpha$ takes the following form:

$$iM_\alpha = W_\alpha\left\{1 - \frac{1}{4}\bar{D}^2\left[\bar{W}^2\big(K + \frac{1}{8}D^2\big(W^2\frac{\partial K}{\partial u}\big)\big)\right]\right\} \qquad (4.19a)$$

$$= W_\alpha\big(1-2\bar{u}\Gamma\big) + W\bar{W}(\cdots) + W^2(\cdots). \qquad (4.19b)$$

Note that in (4.17b) and (4.19b) the ellipsis are quite involved terms that we avoided writing since they will not contribute to $\mathcal{O}_{T^2}$ due to the nilpotency conditions $W_\alpha W_\beta W_\gamma = 0$ and $\bar{W}_{\dot{\alpha}}\bar{W}_{\dot{\beta}}\bar{W}_{\dot{\gamma}} = 0$. In fact, for our purposes, we can further simplify the expressions of $\mathcal{J}_{\alpha\dot{\alpha}}$ and $\mathcal{X}$ to

$$\mathcal{X} = \frac{4\cosh(\gamma)}{3} W^2\bar{u}\big(\Gamma+\bar{\Gamma}-K\big) + W^2\bar{W}(\cdots), \qquad (4.20a)$$

$$\mathcal{J}_{\alpha\dot{\alpha}} = \cosh(\gamma)\left\{-4W_\alpha\bar{W}_{\dot{\alpha}}\big(1-\bar{u}\Gamma-u\bar{\Gamma}\big) + \frac{1}{6}(D_\alpha W^2)(\bar{D}_{\dot{\alpha}}\bar{W}^2)\big(\Gamma+\bar{\Gamma}-K\big)\right\}$$
$$+ W^2\bar{W}(\cdots) + \bar{W}^2 W(\cdots). \qquad (4.20b)$$

Before continuing our analysis, it is worth mentioning that the condition for the general Lagrangian $\mathcal{L}_\Lambda$ in (4.16) to be invariant under electro-magnetic duality transformation is

$$\text{Im}\big\{\Gamma - \bar{u}\Gamma^2\big\} = 0. \qquad (4.21)$$

---

[11]See [67] for a review of $4d$, $\mathcal{N}=1$ old-minimal supergravity in the notation of [41].

The previous self-duality condition was introduced for the first time in [39, 40] where the general theory of $\mathcal{N} = 1$ and $\mathcal{N} = 2$ supersymmetric nonlinear duality invariant systems was developed. Under the condition (4.21), the supercurrent multiplet of the theory $\mathcal{L}_\Lambda$ is also electro-magnetic duality invariant, see [41] for details. The $\mathcal{N} = 1$ supersymmetric Mod-Max theory is a special case of the analysis of [39–41] where the vector multiplet Lagrangian is not required to be analytic. It was in fact proven in [47] that the supersymmetric Born-Infeld-ModMax Lagrangian $\mathcal{L}_{\text{susy}-\gamma\text{BI}}$, eq. (4.11), indeed satisfies (4.21). Exactly as in the non-supersymmetric case, this indicates that any deformation triggered by composite operators defined only in terms of the supercurrent should remain electro-magnetic invariant.

As a next step towards proving (4.12), we use equations (4.20a) and (4.20b) for the FZ superfields to compute $\mathcal{X}\bar{\mathcal{X}}$ and $\mathcal{J}^{\alpha\dot\alpha}\mathcal{J}_{\alpha\dot\alpha}$. The first one can be easily computed to be

$$\mathcal{X}\bar{\mathcal{X}} = \frac{16\cosh^2(\gamma)}{9}W^2\bar{W}^2 u\bar{u}(\Gamma + \bar{\Gamma} - K)^2. \tag{4.22}$$

The expression for $\mathcal{J}^{\alpha\dot\alpha}\mathcal{J}_{\alpha\dot\alpha}$ is more involved. It proves to be

$$\begin{aligned}\mathcal{J}^{\alpha\dot\alpha}\mathcal{J}_{\alpha\dot\alpha} &= 16\cosh^2(\gamma)W^2\bar{W}^2\left\{\left(1 - \bar{u}\Gamma - u\bar{\Gamma}\right)^2 + \frac{1}{9}u\bar{u}\left(\Gamma + \bar{\Gamma} - K\right)^2\right\}\\ &\quad -\frac{4\cosh^2(\gamma)}{3}W^2\bar{W}^2(D^\alpha W_\alpha)^2\left(1 - \bar{u}\Gamma - u\bar{\Gamma}\right)\left(\Gamma + \bar{\Gamma} - K\right).\end{aligned} \tag{4.23}$$

Note that the second line in (4.23) is in principle problematic to prove the flow equation (4.12). In fact, it is clear that the derivative of $\mathcal{L}_{\text{susy}-\text{BI}}$ with respect to $\alpha^2$ is

$$\frac{\partial\mathcal{L}_{\text{susy}-\text{BI}}}{\partial\alpha^2} = -\frac{t^2}{4}\cosh(\gamma)\int d^2\theta d^2\bar{\theta}\,W^2\bar{W}^2\frac{\partial K(u,\bar{u})}{\partial t}, \tag{4.24}$$

where

$$\cosh(\gamma)\frac{\partial K(u,\bar{u})}{\partial t} = \frac{1}{u\bar{u}}\left\{1 - \frac{2t - 2\sinh(g)\sqrt{u\bar{u}} + \cosh(\gamma)(u + \bar{u})}{\sqrt{+4t^2 - 8t\sinh(\gamma)\sqrt{u\bar{u}} + 4t\cosh(\gamma)(u + \bar{u}) + (u - \bar{u})^2}}\right\}. \tag{4.25}$$

Therefore, for the flow (4.12) to hold the operator $\mathcal{O}_{T^2}$ should be a functional of $u$ and $\bar{u}$ only. However, the term in (4.23) that includes the $(D^\alpha W_\alpha)^2$ factor is incompatible with this statement.

The solution to this problem is the same as the one given in [25] for the $\gamma = 0$ case.[12] In fact, it is enough to prove that, for any $\gamma$, the superspace equations of motion derived from the Lagrangian $\mathcal{L}_{\text{susy}-\gamma\text{BI}}$, eq. (4.11), imply the following relation

$$W^2\bar{W}^2(D^\alpha W_\alpha) \equiv 0. \tag{4.26}$$

We refer the reader to Appendix A of [25] for a detailed discussion of this result for a large class of models that include the Lagrangian $\mathcal{L}_\Lambda$ in eq. (4.16), and, in particular, also $\mathcal{L}_{\text{susy}-\gamma\text{BI}}$ in eq. (4.11). Notably, equation (4.26) is equivalent to the fact that the auxiliary field $D \propto D^\alpha W_\alpha|_{\theta=0}$ satisfies an algebraic equation of motion that sets it to zero up to terms at least linear in gaugino fields $\lambda_\alpha \propto W_\alpha|_{\theta=0}$. For the Born-Infeld-like ModMax theory, this fact — directly related to the preservation of supersymmetry on-shell — was also discussed in [47]. Note also that equation (4.26) alone is a weaker condition than imposing the whole set of superfield equations of motion. In fact, imposing (4.26) can be interpreted as only eliminating

---

[12]A similar problem and its solution were also described in the analysis of flow equations of $2d$, $\mathcal{N} = (1,1)$ and $\mathcal{N} = (2,2)$ supersymmetric theories where off-shell supersymmetric multiplets include auxiliary fields [6, 9, 25].

the auxiliary field D from the vector multiplet and removing possible ambiguities of the off-shell description of supersymmetric Born-Infeld-like theories — see e.g. [25, 65, 68–70] for related discussions.

Upon imposing the condition (4.26), it is simple to show that the $\mathcal{O}_{T^2}$ operator (4.7) takes the simple form

$$\mathcal{O}_{T^2} = \cosh^2(\gamma)\, W^2 \bar{W}^2 \Big[ \big(1 - \bar{u}\Gamma - u\bar{\Gamma}\big)^2 - u\bar{u}\big(\Gamma + \bar{\Gamma} - K\big)^2 \Big], \qquad (4.27)$$

and it is only a functional of $u$ and $\bar{u}$. To conclude our analysis and finally show that the flow equation (4.12) is satisfied, it is now enough to compute explicitly (4.27) by using the expression of $K(u,\bar{u})$ (4.15) and the definitions of $\Gamma(u,\bar{u})$ and $\bar{\Gamma}(u,\bar{u})$ in (4.18). A straightforward calculation shows that the right hand side of (4.27) precisely coincides with (4.25) up to a multiplicative factor:

$$\mathcal{O}_{T^2} = -2t^2 \cosh(\gamma)\, \bar{W}^2 W^2 \frac{\partial K(u,\bar{u})}{\partial t}. \qquad (4.28)$$

By comparing (4.28) with (4.24) it follows that the flow equation (4.12) is satisfied. Remarkably, as for the bosonic case, the structure of the supersymmetric flow equation, and its supercurrent-squared operator, proves to be the same for any value of $\gamma$.

## 5 Conclusion

In this work, we have seen that the $T^2$ deformation of the ModMax theory is exactly the known Born-Infeld-type generalization of the ModMax theory. Much like the $\gamma = 0$ case of this statement, which is the fact that the $T^2$ deformation of the free Maxwell theory in $d = 4$ gives the usual Born-Infeld action, the flow can also be recast in $\mathcal{N} = 1$ superspace, so that the supersymmetric extension of the ModMax-BI theory satisfies a supercurrent-squared flow. This ModMax-BI theory therefore belongs to a collection of other interesting theories which satisfy current-squared flows in various numbers of dimensions, such as the Dirac action in $d = 2$ and the usual Born-Infeld action in $d = 4$.

There remain many open questions and directions for future research. First, there is the important conceptual question of what is "special" about theories which satisfy $T^2$ flows. Many of the previously studied examples of such theories, like the Dirac and Born-Infeld Lagrangians, are related to strings and branes. It would be very interesting to understand whether there was a more fundamental reason why current-squared flows generate theories of this type, and to see whether there are other examples of interesting theories that satisfy $T^2$ flows or related differential equations.

One hint which may prove useful in answering this question is the relationship between $T^2$ flows and spontaneously broken symmetries. For instance, the Dirac action which describes the scalar transverse fluctuations of a brane is uniquely fixed by the fact that a brane spontaneously breaks a fraction of the Poincaré symmetry of the ambient space in which it is embedded. The fact that the ordinary $2d$ $T\bar{T}$ deformation generates the Dirac Lagrangian suggests that this flow has some relationship with spontaneously broken symmetries (which are then non-linearly realized). In the supersymmetric context, it is known the the Bagger-Galperin model which represents the supercurrent-squared deformation of a free super-Maxwell theory also possesses an extra non-linearly realized supersymmetry [31, 71]. It was also shown that (Volkov-Akulov) Goldstino actions with non-linearly realised supersymmetry satisfy $T\bar{T}$-like flows in $d = 2$ [9, 11, 72] and $d = 4$ [25]. One would like to sharpen these observations and perhaps understand whether a similar symmetry breaking pattern is relevant for the ModMax-BI theory.

There is a set of related questions concerning scalars (some rudimentary comments concerning $T\overline{T}$-like flows for scalar theories are collected in Appendix B). For the ordinary Born-Infeld theory, it is clear that one can incorporate scalars $X^M$ by promoting $S_{\mathrm{BI}}$ to the Dirac-Born-Infeld action $S_{\mathrm{DBI}}$:

$$S_{\mathrm{DBI}} = -T_p \int d^{p+1}\sigma \sqrt{-\det(g_{\mu\nu} + \alpha F_{\mu\nu})}, \tag{5.1}$$

where $g_{\mu\nu}$ is the induced metric

$$g_{\mu\nu} = G_{MN}\partial_\mu X^M \partial_\nu X^N. \tag{5.2}$$

However, it is less clear how to incorporate scalars into the ModMax-BI theory. One proposal for such a generalization was presented in [73], but it is not obvious that this is the unique modification which includes scalars, nor is it clear how to interpret this Lagrangian from the perspective of string theory or a modification of brane physics. Therefore, one might ask what principle one should use in order to define a "Mod-DBI" theory – that is, a two-parameter family of theories labeled by $\gamma, \lambda$, including both a gauge field and scalars, and which reduces to the ModMax-BI theory when the scalars are set to zero. In particular, given such a family, one could instead ask what happens when the *gauge* sector is set to zero. The result would be a two-parameter family of "ModDirac" theories which reduces to the Dirac Lagrangian when $\gamma = 0$. On the other hand, when $\lambda = 0$, this would yield a new $\gamma$-deformed theory of a scalar which is analogous to the ModMax theory.

One way to probe this question about scalars would be to enhance the amount of supersymmetry. In this work, we have focused on the case of $\mathcal{N} = 1$ in four dimensions and considered theories of a vector superfield strength $W_\alpha$. However, with extended supersymmetry such as $\mathcal{N} = 2$, the supersymmetric completion of the ModMax-BI theory includes additional fields needed to complete the multiplet – see [61] for the $\mathcal{N} = 2$ supersymmetric extension of ModMax. In particular, there is a scalar sector. Given a suitable $\mathcal{N} = 2$ version of the supercurrent-squared deformation, one could attempt to solve the superspace flow equation and then study the dynamics of the scalars in the resulting deformed theory. This gives a potentially different proposal for incorporating scalars into the ModMax-BI theory, which is not obviously related to the proposal of [73]. The study of Volkov-Akulov-Dirac-Born-Infeld actions with extended supersymmetry in various space-time dimensions and their relationship to string theory has received attention in the past. In particular, the standard $\gamma = 0$, Born-Infeld case with extended supersymmetry has been already studied in [39, 40, 74–90]. These works might be a starting point to look for $\mathcal{N} = 2$, ModMax-BI deformations.

Even without scalars, there are at least other two ways in which one could try to generalize these observations relating ModMax-BI to $T^2$ flows.

1. One way is to look for theories of $p$-form field strengths for $p > 2$ which satisfy an appropriate flow equation. It was pointed out in [29] that the $T^2$ deformation of a free 3-form field strength in six dimensions, whose undeformed Lagrangian is proportional to $F_{\mu\nu\rho}F^{\mu\nu\rho}$, does not give a duality-invariant $6d$ analogue of the Born-Infeld theory. However, one could ask whether there is any choice of form rank $p$, dimension $d$, and coefficient $r$ in the operator $O_{T^2}^{[r]}$ in eq. (2.4) for which the deformation of a free $p$-form yields an interesting theory (for instance, with a square root structure). It would also be interesting to consider deformations involving chiral $p$-forms. Since the ModMax theory lifts to a $6d$ PST-like theory of a chiral tensor [46], it is natural to wonder about the relationship between $T^2$ and theories of this kind.

2. Another direction for generalization is to consider non-Abelian gauge theories.[13] The formalism developed in Section 3 does not apply in the non-Abelian case because, for each fixed spacetime trace structure $x_j$, there can be multiple inequivalent ways to perform the traces over gauge indices. An analogue of the master flow equation has not yet been written down in the non-Abelian case, but it is known that the $T\overline{T}$ deformation of free Yang-Mills does not agree with the non-Abelian DBI action in any number of spacetime dimensions. For instance, in $d = 2$ the solution of the $T\overline{T}$ for free Yang-Mills coupled to scalars was discussed in [92]. Even though these deformed theories are no longer related to Born-Infeld in the non-Abelian case, they may still be interesting theories in their own right. It might therefore be worthwhile to consider the behavior of a non-Abelian ModMax-type theory under $T^2$ flows.

A final puzzle, which we have already alluded to before, is the brane interpretation of the ModMax family of theories. For instance, if the ModMax-BI theory exists at the quantum level, then it should have had a string theoretic interpretation as some deformation of the usual Born-Infeld theory on a brane. What deformation does this correspond to? Is there some brane configuration, perhaps with additional fluxes turned on or other string theory ingredients, which would engineer ModMax-BI in the sense that the brane would have a ModMax-BI theory living on its worldvolume? To our knowledge, stringy constructions of ModMax have not appeared yet. We leave this and the preceding interesting questions to future work.

## Acknowledgements

We are grateful to Hongliang Jiang, Sergei Kuzenko, Emmanouil Raptakis, Savdeep Sethi, and Dmitri Sorokin for discussions and correspondence related to this work. We also thank Stephen Ebert, Hao-Yu Sun, and Zhengdi Sun for comments on an early draft of this manuscript. C.F. is supported by U.S. Department of Energy grant DE-SC0009999 and by funds from the University of California. L.S. is supported by a postgraduate scholarship at the University of Queensland. The work of G.T.-M. is supported by the Australian Research Council (ARC) Future Fellowship FT180100353, and by the Capacity Building Package of the University of Queensland.

## A    Proof of Determinant Condition

The goal of this Appendix is to prove that the stress-energy tensor $T_{\mu\nu}$ for any theory of an Abelian gauge field in four spacetime dimensions satisfies

$$\sqrt{\det(T)} = \frac{1}{4}\left(\frac{1}{2}(\operatorname{tr}(T))^2 - \operatorname{tr}\left(T^2\right)\right). \tag{A.1}$$

To see this, we first recall from Section 3.1 that a general Lagrangian for an Abelian field strength $F_{\mu\nu}$ in four dimensions can be written as $\mathcal{L}(x_1, x_2)$ in terms of the two independent scalars $x_1, x_2$, and that the associated stress-energy tensor is (in Euclidean signature)

$$T_{\mu\nu} = \delta_{\mu\nu}\mathcal{L} - 4\frac{\partial\mathcal{L}}{\partial x_1}F_{\mu\nu}^2 - 8\frac{\partial\mathcal{L}}{\partial x_2}F_{\mu\nu}^4. \tag{A.2}$$

---

[13]Another interesting, but very different, connection between $T\overline{T}$ and non-Abelian gauge theory involves a $4d$ version of Chern-Simons as in [91].

At each fixed spacetime location $x$, the stress-energy tensor can therefore be written in components as a $4 \times 4$ matrix of the form

$$T = c_0 \mathbb{I}_4 + c_1 F^2 + c_2 F^4 , \tag{A.3}$$

where $\mathbb{I}_4$ is the identity matrix and the $c_i$ are numbers. We claim that any matrix of the form (A.3), where $F$ is antisymmetric, satisfies (A.1). If the eigenvalues of the antisymmetric matrix $F$ are $\lambda_i$, then the eigenvalues of $T$ are $\hat{\lambda}_i = c_0 + c_1 \lambda_i^2 + c_2 \lambda_i^4$, for $i = 1, \cdots, 4$. The eigenvalues of an antisymmetric matrix are purely imaginary and come in complex conjugate pairs, so we can take $\lambda_3 = \lambda_1^* = -\lambda_1$ and $\lambda_4 = \lambda_2^* = -\lambda_2$. It follows that $\hat{\lambda}_3 = \hat{\lambda}_1$ and $\hat{\lambda}_4 = \hat{\lambda}_2$. Hence, it holds

$$\sqrt{\det(T)} = \sqrt{\hat{\lambda}_1^2 \hat{\lambda}_2^2} = \pm \hat{\lambda}_1 \hat{\lambda}_2 . \tag{A.4}$$

We can take the positive sign on the right side of (A.4) if the stress-energy tensor $T$ is positive definite. On the other hand,

$$\begin{aligned}
\frac{1}{2} (\operatorname{tr}(T))^2 - \operatorname{tr}(T^2) &= \frac{1}{2} (2\hat{\lambda}_1 + 2\hat{\lambda}_2)^2 - (2\hat{\lambda}_1^2 + 2\hat{\lambda}_2^2) \\
&= 4\hat{\lambda}_1 \hat{\lambda}_2 .
\end{aligned} \tag{A.5}$$

Therefore, assuming $T$ is positive definite so that its determinant is positive, one has

$$\sqrt{\det(T)} = \frac{1}{4} \left( \frac{1}{2} (\operatorname{tr}(T))^2 - \operatorname{tr}(T^2) \right) , \tag{A.6}$$

for the stress-energy tensor $T_{\mu\nu}$ of a general Abelian gauge theory in four spacetime dimensions.[14]

A similar result holds in any even number $d = 2k$ of spacetime dimensions.[15] The stress-energy tensor (3.10) for an Abelian gauge theory in any dimension takes the form

$$T = c_0 \mathbb{I}_4 + \sum_{i=1}^{d} c_i F^{2i} , \tag{A.7}$$

and if the eigenvalues of the antisymmetric $d \times d$ matrix $F$ are denoted $\lambda_i$, then the eigenvalues of $T$ are

$$\hat{\lambda}_i = c_0 + \sum_{i=1}^{k} c_i \lambda_i^{2i} . \tag{A.8}$$

Since the eigenvalues again come in complex conjugate pairs, we can choose the first $k$ eigenvalues to be independent and then impose $\lambda_{k+1} = \lambda_1^* = -\lambda_1, \cdots, \lambda_d = \lambda_k^* = -\lambda_k$. This means that

$$\hat{\lambda}_{k+1} = \hat{\lambda}_1, \cdots, \hat{\lambda}_d = \hat{\lambda}_k , \tag{A.9}$$

and if the stress-energy tensor is positive definite, one then has

$$\sqrt{\det(T)} = \sqrt{\hat{\lambda}_1^2 \cdots \hat{\lambda}_k^2} = \hat{\lambda}_1 \cdots \hat{\lambda}_k . \tag{A.10}$$

---

[14]Since $\det(T)$ also satisfies equation (3.16), which applies to any $4 \times 4$ matrix, one could eliminate $\det(T)$ and express this condition as the vanishing of a particular combination of traces of powers of $T$.

[15]In the case of odd spacetime dimension, one must account for the fact that the field strength $F_{\mu\nu}$ has an unpaired zero eigenvalue but the others come in complex-conjugate pairs.

It is an elementary result in the theory of symmetric polynomials, which follows from Newton's identities, that the symmetric polynomial $\hat{\lambda}_1 \cdots \hat{\lambda}_k$ can be expressed in terms of power sums of the form $\hat{\lambda}_1^j + \cdots + \hat{\lambda}_k^j$, which in turn means that (A.10) can be expressed in terms of traces of powers of the matrix $T$. Explicitly, one has

$$\sqrt{\det(T)} = (-1)^k \sum_{\{m_j\}} \left[ \prod_{j=1}^{k} \frac{1}{m_j! j^{m_j}} \left( -\frac{1}{2} \operatorname{tr}(T^j) \right)^{m_j} \right], \tag{A.11}$$

where the sum runs over all collections $\{m_j\}$ of non-negative integers which satisfy the constraint $m_1 + 2m_2 + \cdots + km_k = k$. For instance, in the case of a 6-dimensional Abelian gauge theory, one finds

$$\sqrt{\det(T)} = \frac{1}{6} \left( (\operatorname{tr}(T))^3 - 3 \operatorname{tr}(T^2) \operatorname{tr}(T) + 2 \operatorname{tr}(T^3) \right). \tag{A.12}$$

However, beyond $d = 4$ we see that such combinations are not related to bilinears in stress-energy tensors but rather products involving three or more stress-energy tensor factors. Therefore there does not appear to be any relationship between the combination $\sqrt{\det(T)}$ and any analogue of the $O_{T^2}^{[r]}$ operator for deformations of gauge theories in $d > 4$.

Finally, there has been some interest [11, 64] in $T\overline{T}$-like deformations in higher dimension which involve other powers of the determinant of the stress-energy tensor, such as $[\det(T)]^{1/(d-1)}$. From the analysis of this Appendix, we see that such an operator can never be written in terms of traces of integer powers of the stress-energy tensor when the exponent is smaller than $1/2$, at least in the case of a stress-energy tensor for an Abelian gauge theory. This is because $[\det(T)]^{1/N}$ will involve fractional powers of the eigenvalues $\hat{\lambda}_i$ whenever $N > 2$, whereas polynomials in traces of integer powers of $T$ can produce only integer powers of the $\hat{\lambda}_i$.

# B General $T^2$ Flows for Scalar Theories

In the body of this paper, we have focused on theories whose only physical degree of freedom is an Abelian gauge field, such as the ModMax theory and its ModMax-BI extension. However, as was pointed out in the concluding comments of Section 5, it is natural to wonder about families of theories that involve a gauge field coupled to scalars – such as the Dirac-Born-Infeld (DBI) action – and how such theories interact with $T^2$ flows. The aim of this Appendix is to make some preliminary observations in this direction.

Any theory of a gauge field coupled to scalars must, of course, reduce to a pure gauge theory when the scalar sector is turned off, and must reduce to a scalar theory when the field strength is set to zero. Therefore, if such a coupled theory is driven by a $T^2$ flow, then as a consistency check we know that the pure gauge sector must satisfy a flow equation of the form developed in Section 3 for general gauge theories. A second consistency check is provided by the requirement that the coupled theory satisfy a version of the master flow equation for scalar fields when the gauge sector is turned off. As a first step towards understanding flows for coupled theories, one would like to repeat the general analysis of Section 3 in the case of a scalar field to obtain a second boundary condition for coupled flows. In this Appendix we will complete such a first step.

For simplicity, we will restrict our attention to theories of a single scalar field $\phi$. We first make some general comments which apply in any spacetime dimension $d$.

## B.1 Master Flow Equation for Scalars

In this subsection, we would like to obtain a general flow equation for a Lagrangian $\mathcal{L}(\phi)$ for a single scalar field $\phi$ deformed by some Lorentz scalar constructed from the stress tensor $T_{\mu\nu}$. Our discussion will parallel the derivation of the master flow equation for theories involving a gauge field in Section 3.1, although the scalar case is considerably simpler, which will motivate us to consider more general deformations.

We first note that any Lorentz invariant scalar that can be constructed from $\phi$ with one derivative per field is a function of the combination

$$x = \partial^\mu \phi \, \partial_\mu \phi \,. \tag{B.1}$$

Thus a general Lagrangian for a theory of a single scalar field can be written as $\mathcal{L} = \mathcal{L}(x)$. The Hilbert stress tensor corresponding to this Lagrangian is

$$
\begin{aligned}
T_{\mu\nu} &= \delta_{\mu\nu}\mathcal{L} - 2\frac{\partial L}{\partial x} \cdot \frac{\delta x}{\delta g^{\mu\nu}}\bigg|_{g=\delta} \\
&= \delta_{\mu\nu}\mathcal{L} - 2\frac{\partial \mathcal{L}}{\partial x} \cdot \partial_\mu \phi \, \partial_\nu \phi \,.
\end{aligned}
\tag{B.2}
$$

First we will describe the independent scalars that can be constructed from this stress tensor. The determinant is especially simple, since in a component basis at a particular spacetime point $p$, $T_{\mu\nu}(p)$ is written as a linear combination of the identity matrix and the matrix $M_{\mu\nu} = \partial_\mu \phi \, \partial_\nu \phi$. Because $M_{\mu\nu}$ is the outer product of a vector with itself, it is rank one and has only a single non-zero eigenvalue. To make this very explicit, if $\boldsymbol{v}$ is the vector obtained by writing the components of $\partial_\mu \phi$ in a given basis at a fixed spacetime location $p$, then

$$M = \boldsymbol{v} \otimes \boldsymbol{v} \,. \tag{B.3}$$

It is an elementary fact from linear algebra that a $d \times d$ matrix $M$ which can be written as the outer product $\boldsymbol{v} \otimes \boldsymbol{v}$ has one eigenvalue equal to $|\boldsymbol{v}|^2$ and $d-1$ eigenvalues equal to zero. This will allow us to easily evaluate the eigenvalues of the matrix $T$, which can be written in components at a fixed point $p$ as

$$T = c_0 \mathbb{I} + c_1 \boldsymbol{v} \otimes \boldsymbol{v} \,. \tag{B.4}$$

Here $c_0$ and $c_1$ are numbers which depend on the value of the Lagrangian and its derivatives at the point $p$, but which can be treated as constants for this local analysis. Owing to the outer product structure of the second term, the matrix $T$ has $d-1$ eigenvalues equal to $c_0$ and a single eigenvalue equal to $c_0 + c_1 x$, where $x = |\boldsymbol{v}|^2$ is the value of $\partial^\mu \phi \, \partial_\mu \phi$ at the point $p$. This means that, in an arbitrary number $d$ of spacetime dimensions, the determinant of $T$ is

$$\det(T) = \mathcal{L}^{d-1} \cdot \left(\mathcal{L} - 2x\frac{\partial \mathcal{L}}{\partial x}\right) \,. \tag{B.5}$$

The other scalars that can be constructed from the stress tensor are traces of the form $y_k = \operatorname{tr}(T^k)$. Using our result for the eigenvalues of $T$ above, it is easy to write down a general formula for such traces:

$$y_k = \operatorname{tr}(T^k) = (d-1)\mathcal{L}^k + \left(\mathcal{L} - 2x\frac{\partial \mathcal{L}}{\partial x}\right)^k \,. \tag{B.6}$$

From the Cayley-Hamilton theorem, we know that the determinant $\det(T)$ can be written as a polynomial in the traces $y_k$. Furthermore, any trace $y_k$ for $k > d$ also satisfies a constraint which relates it to the lower traces $y_j$ for $j = 1, \cdots, k$. Thus a general flow equation for the

Lagrangian driven by a deforming operator $O$ which is a scalar built from the stress tensor $T_{\mu\nu}$ is

$$\frac{\partial \mathcal{L}}{\partial \lambda} = O(y_1, \cdots, y_d). \tag{B.7}$$

For instance, the main operator of interest in this manuscript has been $O_{T^2}^{[r]}$, which can be written as

$$O_{T^2}^{[r]}(y_1, y_2) = y_2 - r y_1^2. \tag{B.8}$$

We can use the expressions (B.6) for the $y_k$ to write a master flow equations for scalar theories in $d$ dimensions. One has

$$y_1 = \text{tr}(T) = \mathcal{L}d - 2x \frac{\partial \mathcal{L}}{\partial x}, \qquad y_2 = \text{tr}(T^2) = \mathcal{L}^2 d - 4\mathcal{L}x\frac{\partial \mathcal{L}}{\partial x} + 4x^2 \left(\frac{\partial \mathcal{L}}{\partial x}\right)^2. \tag{B.9}$$

Therefore, a general flow driven by the operator of $\mathcal{O}_{T^2}^{[r]}$ of equation (B.8) is described by the differential equation

$$\frac{\partial \mathcal{L}}{\partial \lambda} = d(1 - rd)\mathcal{L}^2 + 4(rd - 1)\mathcal{L}x\frac{\partial \mathcal{L}}{\partial x} + 4(1 - r)x^2 \left(\frac{\partial \mathcal{L}}{\partial x}\right)^2. \tag{B.10}$$

This is the analogue of the master flow equation (3.12), but now for theories involving a single scalar rather than a gauge field. However, we note that in $d > 2$ there are more scalar invariants associated with the stress tensor and one may therefore consider more general deforming operators. For instance, we can obtain another operator with the same mass dimension as $O_{T^2}^{[r]}$ by taking a square root of traces involving products of four stress tensors. We adopt the notation $O_{\sqrt{T^4}}^{[r_i]}$ for such an operator, which depends on three coefficients $r_1, r_2, r_3$ as

$$O_{\sqrt{T^4}}^{[r_i]}(y_1, \cdots, y_4) = \sqrt{y_1^4 + r_1 y_1^2 y_2 + r_2 y_2^2 + r_3 y_4}. \tag{B.11}$$

We will see below that, in $d = 4$ spacetime dimensions, deforming the Lagrangian for a scalar theory by $\sqrt{\det(T)}$ is classically equivalent to deforming by an operator $O_{\sqrt{T^4}}^{[r_i]}$ for some choice of the constants $r_i$.

Before returning to the study of these deformations by more general $O(y_1, \cdots, y_d)$ operators, we will first undertake an analysis of flows by the usual $O_{T^2}^{[r]}$ for scalar theories in arbitrary dimension.

## B.2 General Analysis of $O_{T^2}^{[r]}$ Flows

Here we will focus on the master flow equation (B.10) for scalar field theories deformed by $O_{T^2}^{[r]}$. Since it is known that this flow equation has a solution of Nambu-Goto type in $d = 2$, one might ask whether there are other solutions involving such a square root structure in $d > 2$. This is the scalar analogue of the question of whether the Born-Infeld action (or its ModMax-BI extension) emerges as a $T^2$ flow in any dimension other than $d = 4$, to which we have seen that the answer is no.

We first note that, on dimensional grounds, any Lagrangian which depends only on $\lambda$ and $x$ can be written as

$$\mathcal{L}(\lambda, x) = \frac{1}{\lambda} f(\lambda x), \tag{B.12}$$

where we define $\xi = \lambda x$ as the dimensionless argument of the function $f$. This parameterization reduces the partial differential equation (B.10) to an ordinary differential equation for $f(\xi)$, namely

$$4(r-1)\xi^2 \big(f'(\xi)\big)^2 + \xi f'(\xi) - \big(1 + 4(rd-1)\xi f'(\xi)\big)f(\xi) + d(rd-1)(f(\xi))^2 = 0. \quad \text{(B.13)}$$

This is a quadratic equation in the quantity $f'(\xi)$ which can be solved to give

$$f'(\xi) = \frac{1}{8\xi(r-1)}\Big(4(rd-1)f(\xi) + \sqrt{8f(\xi)(2(d-1)(rd-1)f(\xi) - rd + 2r - 1) + 1} - 1\Big), \quad \text{(B.14)}$$

where we have chosen the root consistent with $f'(\xi)$ being finite as $\xi \to 0$ with $f(0) = 0$. One can separate this differential equation as

$$\int_{f(\xi_0)}^{f(\xi)} \frac{df}{4(rd-1)f + \sqrt{8f(2(d-1)(rd-1)f - rd + 2r - 1) + 1} - 1} = \left[\frac{\log(\xi')}{8(r-1)}\right]_{\xi'=\xi_0}^{\xi'=\xi}. \quad \text{(B.15)}$$

The integral on the left side of (B.15) is quite involved but can be evaluated in closed form in terms of inverse hyperbolic trigonometric functions using Mathematica. The result is not especially illuminating so we do not show it here; we include this integral expression for the solution only to emphasize that, for general $r$ and $d$, the solution for a scalar deformed by $O_{T^2}^{[r]}$ is a fairly complicated implicitly defined function which is structurally similar to the result of $T\overline{T}$ deforming $2d$ Yang-Mills theory coupled to scalars [92] or deforming $2d$ Born-Infeld theory [30].

The implicit expression (B.15) simplifies for particular choices of the coefficient $r$. For instance, when $r = 1$, the quadratic equation for $f'(\xi)$ has the much simpler solution

$$f'(\xi) = \frac{f(\xi)(-1 + (d-1)f(\xi)d)}{\xi(-1 + 4(d-1))f(\xi)}, \quad \text{(B.16)}$$

which can be integrated to yield the implicit equation

$$\log(f) + \frac{4-d}{d}\log\big(1 + f(d - d^2)\big) = \log(\xi). \quad \text{(B.17)}$$

This equation is transcendental for generic $d$, but when $d = 2$, we see that the left side collapses to $\log(f) + \log(1 - 2f)$ and the solution is

$$f(\xi) = \frac{1}{4}\Big(1 - \sqrt{1 - 8\xi}\Big). \quad \text{(B.18)}$$

This is the familar result that the ordinary $T\overline{T}$ deformation applied to a free scalar seed theory in $d = 2$ yields the Nambu-Goto Lagrangian.

Another choice for which the implicit expression simplifies is $r = \frac{1}{d}$, which gives

$$f'(\xi) = \frac{d - \sqrt{d^3 - 16d^2(d-1)f(\xi)}}{8(d-1)\xi}. \quad \text{(B.19)}$$

This differential equation has a solution which can be written in terms of the product logarithm (Lambert $W$ function), but no solution of square-root type.

Given the complexity of the general implicit solution (B.15), and the observation that miraculous simplifications were needed in order to obtain the Nambu-Goto action as a solution with $d = 2$ and $r = 1$, one might suspect that this is the *only* choice of the parameters $r, d$ for which a square-root solution exists. This is easy to verify; we first make an ansatz of the form

$$f(\xi) = \frac{1}{a}\Big(1 - \sqrt{1 - 2a\xi}\Big), \quad \text{(B.20)}$$

where $a$ is some constant. Substituting this ansatz into the ordinary differential equation (B.13) and expanding to second order in $\xi$ yields the constraint

$$a = -8r - 2rd^2 + d(2 + 8r).$$
(B.21)

Using this value of $a$ in the differential equation and expanding to third order in $\xi$ gives the condition

$$(-2 + d)(-4r + d^3 r^2 - 2d^2 r(1 + 2r) + d(1 + 2r)^2).$$
(B.22)

This equation is satisfied if either $d = 2$ or if $r$ takes one of the values

$$r = \frac{1}{d}, \qquad r = \frac{d}{(2 - d)^2}.$$
(B.23)

We handle each of these cases separately.

1. $d = 2$. In this case, plugging the results for $a$ and $d$ back into the flow equation gives an equation which is satisfied if and only if $r = 1$. This reduces to the known case.

2. $r = \frac{1}{d}$. Substitution into (B.13) and expansion to order $\xi^4$ yields the constraint $d = 1$. We reject this since we are interested in $T\overline{T}$ deformations of field theories ($d \geq 2$) rather than quantum mechanics; in one spacetime dimension, the expression $\mathcal{O}_{T^2}^{[r]}$ trivializes because the only component of the "stress tensor" is the Hamiltonian.[16]

3. $r = \frac{d}{(2-d)^2}$. Replacement of $r$ with this value in the flow equation then gives two constraints: $d^2 - 4d + 4 = 0$ and $d^2 - 5d + 4 = 0$. The first condition requires $d = 2$ but the second requires either $d = 1$ or $d = 4$. Thus the two equations cannot be simultaneously satisfied and this choice is inconsistent.

This completes our check that a deformation of a free scalar theory by $O_{T^2}^{[r]}$ only produces the Nambu-Goto Lagrangian as a solution in the single case $d = 2, r = 1$.

Given this conclusion, one is tempted to consider deformations by other Lorentz scalars constructed from the stress tensor. For instance, in $d > 2$ dimensions one can deform the Lagrangian by a power of the determinant of the stress tensor, or by some function of the higher independent trace structures $y_k$ defined in (B.6). We next turn to an investigation of some other deformations of this type in $d = 4$.

## B.3 Other Stress Tensor Flows in $d = 4$

Another deformation constructed from $T_{\mu\nu}$ is $\sqrt{\det(T)}$. We note that this combination agreed with $O_{T^2}^{[r]}$ in the case of $4d$ gauge theory (up to overall scaling), but the two objects disagree for a scalar. In particular, it is no longer true that the eigenvalues of $T_{\mu\nu}$ come in pairs of equal $\hat{\lambda}_i$ since the symmetric tensor $\partial_\mu \phi \partial_\nu \phi$ cannot be written as the square of an antisymmetric tensor in the way that $F_{\mu\nu}^2$ could. Rather, in $d = 4$, the determinant $\det(T)$ is instead equal to $\mathcal{L}^3 \cdot \left( \mathcal{L} - 2x \frac{\partial \mathcal{L}}{\partial x} \right)$ as we saw in (B.5).

The combination $\sqrt{\det(T)}$ is one member of the general class of deformations (B.7) by some function of the traces $y_i = \text{tr}(T^i)$. In particular, since any $4 \times 4$ matrix satisfies (3.16) regardless of its symmetry properties, we have

$$\det(T) = \frac{1}{24} \left( (\text{tr } T)^4 - 6\,\text{tr}(T^2)(\text{tr } T)^2 + 3\left(\text{tr } T^2\right)^2 + 8\,\text{tr}(T)\,\text{tr}(T^3) - 6\,\text{tr}(T^4) \right).$$
(B.24)

---

[16]Although we will not consider this case in the present work, see [93–95] for observations on $T\overline{T}$-like deformations in $(0 + 1)$-dimensional systems.

To construct other deformations from the $y_i$, it will be convenient to record explicit expressions for these traces:

$$
\begin{aligned}
\mathrm{tr}(T) &= 4\mathcal{L} - 2x\frac{\partial \mathcal{L}}{\partial x}, \\
\mathrm{tr}(T^2) &= 4\mathcal{L}^2 - 4x\mathcal{L}\frac{\partial \mathcal{L}}{\partial x} + 4x^2\left(\frac{\partial \mathcal{L}}{\partial x}\right)^2, \\
\mathrm{tr}(T^3) &= 4\mathcal{L}^3 - 6x\mathcal{L}^2\frac{\partial \mathcal{L}}{\partial x} + 12x^2\mathcal{L}\left(\frac{\partial \mathcal{L}}{\partial x}\right)^2 - 8x^3\left(\frac{\partial \mathcal{L}}{\partial x}\right)^3, \\
\mathrm{tr}(T^4) &= 4\mathcal{L}^4 - 8x\mathcal{L}^3\frac{\partial \mathcal{L}}{\partial x} + 24x^2\mathcal{L}^2\left(\frac{\partial \mathcal{L}}{\partial x}\right)^2 - 32x^3\mathcal{L}\left(\frac{\partial \mathcal{L}}{\partial x}\right)^3 + 16x^4\left(\frac{\partial \mathcal{L}}{\partial x}\right)^4.
\end{aligned}
\tag{B.25}
$$

As a check, plugging these trace expressions into (B.24) gives

$$
\det(T) = \mathcal{L}^4 - 2x\mathcal{L}^3\frac{\partial \mathcal{L}}{\partial x},
\tag{B.26}
$$

which matches (B.5). Therefore, a flow equation of the form

$$
\frac{\partial \mathcal{L}}{\partial \lambda} = \sqrt{\det(T^{(\lambda)})}
\tag{B.27}
$$

is equivalent to the differential equation

$$
\frac{\partial \mathcal{L}}{\partial \lambda} = \sqrt{\mathcal{L}^4 - 2x\mathcal{L}^3\frac{\partial \mathcal{L}}{\partial x}}.
\tag{B.28}
$$

We also see that (B.28) is one example of the class of deformations driven by the operators $O^{[r_i]}_{\sqrt{T^4}}$ defined in (B.11), where the coefficients $r_i$ are determined by (B.24).

As written, this flow equation is unsuitable because the argument of the square root need not be positive. For instance, consider the leading order correction in $\lambda$ around a free theory of the form $\mathcal{L}_0 = cx$ where $x$ is some constant. Then the operator on the right side of (B.28) is

$$
\sqrt{\mathcal{L}_0^4 - 2x\mathcal{L}_0^3\frac{\partial \mathcal{L}_0}{\partial x}} = \sqrt{-c^4x^4}.
\tag{B.29}
$$

This is always a purely imaginary correction for any real value of the constant $c$ and the kinetic term $x = \partial^\mu \phi \partial_\mu \phi$. To obtain a real deformation at leading order, one should instead consider the flow

$$
\begin{aligned}
\frac{\partial \mathcal{L}}{\partial \lambda} &= \sqrt{-\det(T^{(\lambda)})} \\
&= \sqrt{2x\mathcal{L}^3\frac{\partial \mathcal{L}}{\partial x} - \mathcal{L}^4}.
\end{aligned}
\tag{B.30}
$$

This differential equation has the solution

$$
\mathcal{L}(\lambda, x) = \frac{x}{\sqrt{1 - 2\lambda x}}.
\tag{B.31}
$$

Solutions of this form for flow equations driven by a power of $\det(T)$ were obtained in [11] using a different strategy.

One could ask whether there is a flow by some other $O(y_i)$ that reproduces the usual Dirac action. Consider the combination

$$O_{T^4}(y_i) = (\operatorname{tr} T)^4 - \frac{1}{3}\operatorname{tr}(T^2)(\operatorname{tr} T)^2 + \frac{1}{3}(\operatorname{tr} T^2)^2 - \operatorname{tr}(T^4)$$

$$= y_1^4 - \frac{1}{3}y_1^2 y_2 + \frac{1}{3}y_2^2 - y_4. \tag{B.32}$$

The square root of this object again drives a flow by an operator of the form $O_{\sqrt{T^4}}^{[r_i]}$, but with a different choice of the coefficients $r_i$ than the one which gives $\sqrt{\det(T)}$. Plugging in the expressions (B.25) for the traces gives

$$O_{T^4} = \left(\mathcal{L}^2 - 2\mathcal{L}x\frac{\partial \mathcal{L}}{\partial x}\right)^2, \tag{B.33}$$

and therefore the flow equation

$$\frac{\partial \mathcal{L}}{\partial \lambda} = \sqrt{O_{T^4}}, \tag{B.34}$$

where we take the positive root, gives

$$\frac{\partial \mathcal{L}}{\partial \lambda} = \mathcal{L}^2 - 2\mathcal{L}x\frac{\partial \mathcal{L}}{\partial x}. \tag{B.35}$$

This differential equation has the solution

$$\mathcal{L}(\lambda, x) = \frac{1}{2\lambda}\left(1 - \sqrt{1 + 4\lambda x}\right). \tag{B.36}$$

Therefore, it is possible to obtain the Dirac Lagrangian as the solution to a stress tensor flow in $d = 4$, although one must use a different deformation $O_{\sqrt{T^4}}^{[r_i]}$ with a special choice of coefficients $r_i$, and it is not clear how one would motivate this particular combination.

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
