# Peer review of "On Current-Squared Flows and ModMax Theories"

_SciPost Physics, doi:SciPost Phys. 13, 012 (2022)_

## Round 2 · Referee Report · Anonymous · 2022-5-13

Strengths
In this manuscript the authors firstly show that the recently introduced ModMax electrodynamics and its
Born-Infeld like generalization, which can be called ModBI, are related by a 4d generalization of so-called
$T\bar{T}$ deformation of 2d field theories which were intensively studied both in classical and quantum regime. To be more precise, it was shown that the ModBI Lagrangian satisfies the flow equation with right-hand side defined by the difference of the square and square of its half-trace. The boundary condition for this equation is given by ModMax Lagrangian.
It is also shown that the superfield generalization of ModBI also obeys the generalized flow equation in which supersymmetric ModMax provides a kind of initial condition. To this end the superfield approach a la Bagger-Galperin is used.
Weaknesses
Also a search for possible multidimensional counterparts of ModBI which obey some flow equations was performed, although in this case the obtained negative result can be not final due to the use of specific ansatze.
Report
In this manuscript the authors firstly show that the recently introduced ModMax electrodynamics and its
Born-Infeld like generalization, which can be called ModBI, are related by a 4d generalization of so-called
$T\bar{T}$ deformation of 2d field theories which were intensively studied both in classical and quantum regime. To be more precise, it was shown that the ModBI Lagrangian satisfies the flow equation with right-hand side defined by the difference of the square and square of its half-trace. The boundary condition for this equation is given by ModMax Lagrangian.
It is also shown that the superfield generalization of ModBI also obeys the generalized flow equation in which supersymmetric ModMax provides a kind of initial condition. To this end the superfield approach a la Bagger-Galperin is used.
Also a search for possible multidimensional counterparts of ModBI which obey some flow equations was performed, although in this case the obtained negative result can be not final due to the use of specific ansatze.
To conclude, this manuscript contains new interesting results, meets criteria of SciPost, including 'expectation n3', and I recommend it for publication.

---

## Round 2 · Referee Report · Anonymous · 2022-6-21

Report
The authors study the Born-Infeld generalization of the modMax theory (they denoted it as modMax-BI) from the point of view of generalized TTbar deformations. They find that indeed the modMax-BI theory can be obtained from the modMax theory, by a flow equation triggered by an operator very similar to the determinant of the energy-momentum tensor. They also show that their result does not generalize to other dimensions, although their argument makes use of a specific ansatz and, thus, is not conclusive. Finally, they successfully extend their result to the supersymmetric version of the modMax theory, where the deforming operator is now a supercurrent squared.
In conclusion, the article is well written and careful in its considerations and results. It contains interesting novel results.
I warmly encourage its publication in sci-post.

---

## Editorial Decision

published